# A phosphorylated transcription factor regulates sterol biosynthesis in *Fusarium graminearum*

Zunyong Liu[1,2], Yunqing Jian[1,2], Yun Chen [1,2], H. Corby Kistler [3], Ping He[4], Zhonghua Ma [1,2] & Yanni Yin[1,2]

Sterol biosynthesis is controlled by transcription factor SREBP in many eukaryotes. Here, we show that SREBP orthologs are not involved in the regulation of sterol biosynthesis in *Fusarium graminearum*, a fungal pathogen of cereal crops worldwide. Instead, sterol production is controlled in this organism by a different transcription factor, FgSR, that forms a homodimer and binds to a 16-bp *cis*-element of its target gene promoters containing two conserved CGAA repeat sequences. FgSR is phosphorylated by the MAP kinase FgHog1, and the phosphorylated FgSR interacts with the chromatin remodeling complex SWI/SNF at the target genes, leading to enhanced transcription. Interestingly, FgSR orthologs exist only in *Sordariomycetes* and *Leotiomycetes* fungi. Additionally, FgSR controls virulence mainly via modulating deoxynivalenol biosynthesis and responses to phytoalexin.

[1] State Key Laboratory of Rice Biology, Zhejiang University, 866 Yuhangtang Road, Hangzhou 310058, China. [2] Institute of Biotechnology, Key Laboratory of Molecular Biology of Crop Pathogens and Insects, Zhejiang University, 866 Yuhangtang Road, Hangzhou 310058, China. [3] United States Department of Agriculture, Agricultural Research Service, 1551 Lindig Street, St. Paul, MN 55108, USA. [4] Department of Biochemistry and Biophysics, Texas A&M University, College Station, TX 77843, USA. Correspondence and requests for materials should be addressed to Z.M. (email: zhma@zju.edu.cn) or to Y.Y. (email: ynyin@zju.edu.cn)

Sterols are essential membrane lipid components, and modulate membrane fluidity, stability, permeability, aerobic metabolism, cell cycle, and the activities of membrane-bound enzymes in most eukaryotic organisms[1–3]. Cholesterol is the major sterol of cell membranes in mammals, while ergosterol is the major sterol of fungal cell membranes. In human, excess cholesterol forms solid crystals that kill cells or deposit in arteries to initiate atherosclerosis, while depleted cholesterol is a risk factor for primary intracerebral hemorrhage[4]. In *Candida albicans* and *Saccharomyces cerevisiae*, ergosterol starvation leads to vacuolar alkalinization by inhibiting the activity of V-ATPase[5]. Hence, the regulation of sterol homeostasis is particularly important for various cellular processes.

To date, two types of sterol regulators, including *Homo sapiens* SREBP (sterol regulatory element binding protein) and *S. cerevisiae* Upc2 (sterol uptake control), have been identified. Regulatory mechanisms of sterol biosynthesis mediated by these regulators have been well characterized. In mammals, the SREBP precursor containing a typical helix-loop-helix domain and two transmembrane domains (TM) is an endoplasmic reticulum (ER) associated transcription factor, and mediates feedback regulation of cholesterol homeostasis through activation by SREBP cleavage activating protein (SCAP)[6]. SCAP possesses eight transmembrane segments in its N terminus that are defined as sterol sensing domain, and multiple WD repeat domains that mediate binding to SREBP in its C-terminus. Under sterol-replete conditions, SCAP directly binds to cholesterol that promotes its interaction with the ER-resident protein Insig (insulin-induced gene) and ultimately allowing for retention of SCAP-SREBP in the ER. However in sterol-depleted cells, the binding of SCAP to Insig is disrupted, and consequently SCAP will then escort SREBP from ER to the Golgi where SREBP is sequentially processed by Site 1 and Site 2 proteases (S1P and S2P) to release the N-terminal transcription factor domain of SREBP from the membrane[4,6,7]. The released SREBP is transported into the nucleus as a dimer by importin β, and subsequently binds to the promoters of target genes involved in cholesterol metabolism[8]. In *Aspergillus fumigatus*, *Cryptococcus neoformans* and *Schizosaccharomyces pombe*, the regulatory mechanism mediated by SREBP orthologs (also named SrbA or Sre1/2) is similar to that in mammals, although some components, such as Insig, SCAP or S1/2P, are not required for cleavage of the SREBPs[9–11].

In *Sacharomycotina* yeasts, ergosterol biosynthesis is mainly modulated by transcription factor Upc2[12,13]. In *S. cerevisiae*, Upc2 contains an N terminal zinc finger domain, and a typical C terminal ligand binding domain which can bind to ergosterol directly to sense the cellular ergosterol level, and subsequently modulates ergosterol biosynthesis[14]. When bound with ergosterol, Upc2 forms an inactive conformation, which inhibits its nuclear localization sequence (NLS) function, and thus it is retained in the cytosol[14]. Under ergosterol-deprived conditions, Upc2 dissociates from ergosterol, and is translocated into the nucleus for transcriptional activation of ergosterol-related genes[14]. Mutation in the C terminus of Upc2 abolishes the sterol binding and increases its transcriptional activity, which leads to resistance to sterol biosynthesis inhibitors (SBIs)[12,14–16]. SBIs inhibit activity of cytochrome P450, leading to decreased ergosterol production and increased toxic intermediates catalyzed by ERG3[17].

*Fusarium graminearum* is the major causal agent of Fusarium head blight (FHB), which is a devastating disease of cereal crops worldwide[18]. In addition to the yield loss caused by the disease, mycotoxins deoxynivalenol (DON) and its derivatives, produced by *F. graminearum* in infested grains, represent a serious threat to human and animal health[19,20]. In the field, management of FHB is mainly dependent upon fungicide application due to the lack of highly resistant plant cultivars[21]. Sterol synthesis inhibitors (azole

drugs) have been applied for the control of FHB for more than 30 years[17,22], yet highly azole-resistant *F. graminearum* strains have not been detected in nature[22,23], in contrast to resistance that arises in other pathogenic fungi, including *Mycosphaerella graminicola*, *Blumeria graminis*, *A. fumigatus*, and *C. albicans*[17,23–25]. Thus, we are interested in exploring the regulatory mechanism of ergosterol biosynthesis in *F. graminearum*.

Here, we show that deletion of the SREBP or Upc2 orthologs in *F. graminearum* does not lead to a change in azole sensitivity. We screen a *F. graminearum* mutant library and find a potential transcription factor FgSR that regulates sterol biosynthesis. We further confirm that FgSR binds to the promoters of several ergosterol biosynthesis genes. Surprisingly, the localization of FgSR is not altered under ergosterol-depleted conditions, which is different from what is known for Upc2 in budding yeast. FgSR is subject to phosphorylation by a mitogen-activated protein (MAP) kinase FgHog1, and phosphorylated FgSR interacts with chromatin-remodeling complex SWI/SNF. FgSR orthologs exist only in *Sordariomycetes* and *Leotiomycetes* fungi. Our study therefore identifies a novel regulatory mechanism of sterol biosynthesis, and provides new cues for management of this devastating pathogen.

## Results

**Identification of a transcription factor regulating ergosterol biosynthesis in *F. graminearum*.** To explore the regulatory mechanism of sterol biosynthesis and homeostasis in *F. graminearum*, we first determined whether transcription factors SREBP and Upc2 homologs regulate sterol biosynthesis in this fungus. In silico analyses showed that the *F. graminearum* genome contains two SREBP orthologs (FgSre1 and FgSre2) and four Upc2 orthologs (FgUpc2A, FgUpc2B, FgUpc2C, and FgUpc2D) (Supplementary Fig. 1a). SREBP orthologs contain a helix-loop-helix, a transmembrane, and a DUF2014 domain. Upc2 orthologs possess a Zn(II)2-Cys6 zinc finger domain and a fungal special transcription factor domain (Supplementary Fig. 1a). We constructed single and double deletion mutants for SREBP and Upc2 orthologs, single deletion mutants for SREBP's partners and three Ndt80 orthologs that have been found to regulate sterol biosynthesis in various eukaryotic organisms[6,9,10,14,26,27]. Fungicide sensitivity determination showed that deletion any one or two of these genes did not alter the susceptibility of *F. graminearum* to the azole compound tebuconazole (Supplementary Fig. 1b). Moreover, the transcription of *FgCYP51A* in the above mutants was similar to that in the wild type, with or without tebuconazole treatment (Supplementary Fig. 1c). In addition, the transcription of each SREBP or Upc2 ortholog was not altered by tebuconazole treatment or deletion of another SREBP or Upc2 ortholog (Supplementary Fig. 1d). These results indicate that sterol biosynthesis in *F. graminearum* may be regulated by an unknown transcription factor other than SREBP and Upc2 orthologs.

Next, we screened for sensitivity to tebuconazole for more than 1000 gene deletion mutants of *F. graminearum* constructed in our laboratory, and found that the FgSR (the locus FGSG_01176) mutant ΔFgSR exhibited markedly increased sensitivity to this compound. Further phenotypic characterization showed that ΔFgSR displayed reduced mycelial growth on potato dextrose agar (PDA), minimal medium (MM), and complete medium (CM) (Fig. 1a), and greatly increased sensitivity to azole compounds, but not to iprodione and fludioxonil that target the high osmolarity glycerol (HOG) pathway (Fig.1b). Given that ergosterol is the major sterol of fungal cell membranes, we also determined the amount of ergosterol in ΔFgSR. As shown in Fig. 1c, the ergosterol content was reduced by 58.06% in ΔFgSR compared to that in the wild type. Domain analysis indicates that

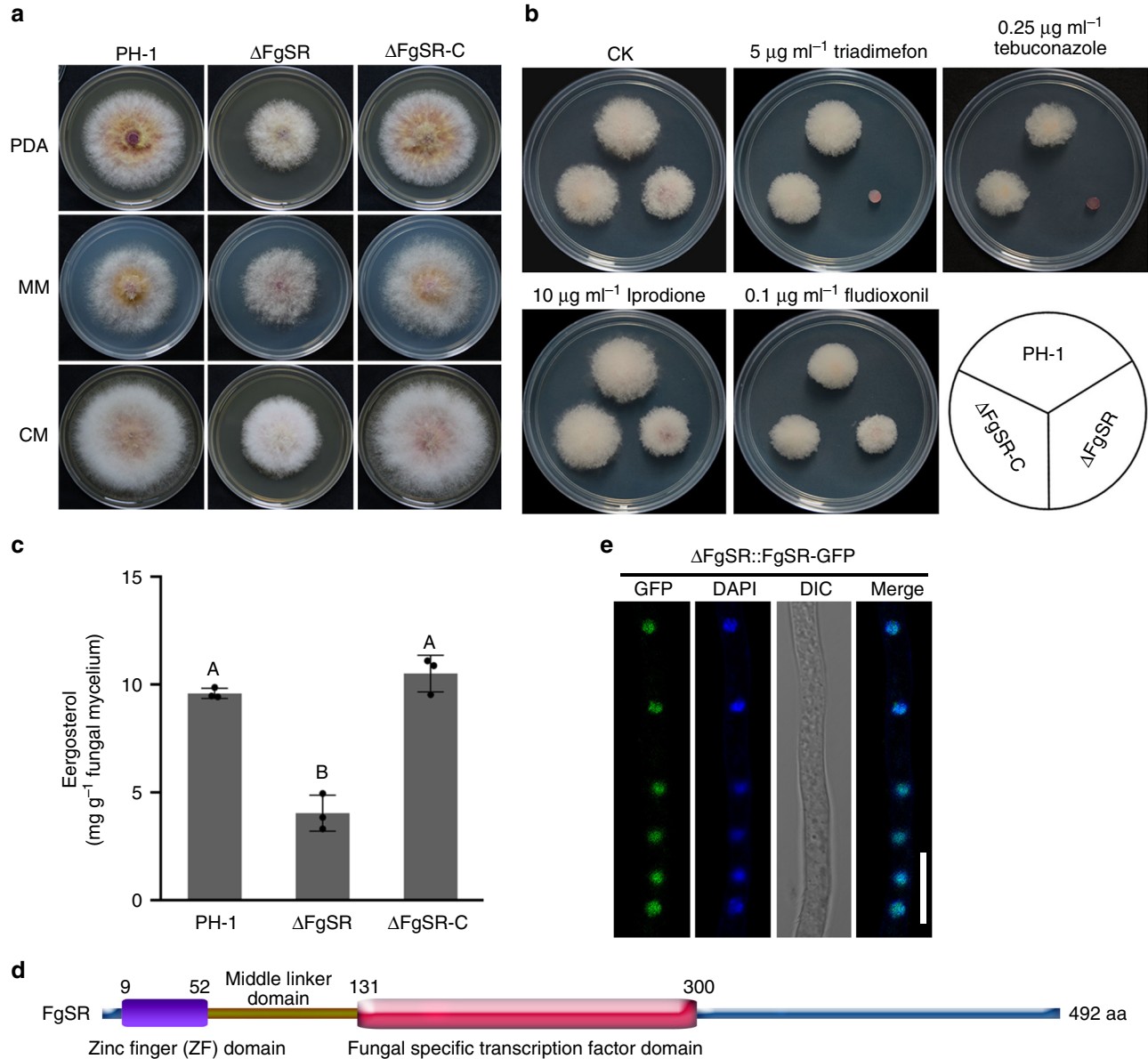

**Fig. 1** Transcription factor FgSR regulates ergosterol biosynthesis in *F. graminearum*. **a** Comparisons in colony morphology among the wild-type PH-1, ΔFgSR and the complemented transformant ΔFgSR-C grown on PDA, CM, or MM. **b** ΔFgSR displayed increased sensitivity to azole compounds but not to the compounds targeting the high osmolarity glycerol (HOG) pathway. A 5-mm mycelial plug of each strain was inoculated on PDA alone or supplemented with 5 μg ml⁻¹ tridiamenfon, 0.25 μg ml⁻¹ tebuconazole, 10 μg ml⁻¹ iprodione or 0.1 μg ml⁻¹ fludioxonil, and then incubated at 25 °C for 2 days. **c** Relative abundance of ergosterol in each strain after growth in YEPD for 36 h. Data presented are the mean ± s.d. ($n = 3$). Bars followed by the same letter are not significantly different according to a Fisher's least significant difference (LSD) at $P = 0.01$. **d** The FgSR amino acid sequence shows a typical Zn(II)2-Cys6 zinc finger domain and a fungal specific transcription factor domain identified using the SMART protein database (http://smart.embl-heidelberg.de) and the NBCI protein database (https://blast.ncbi.nlm.nih.gov/Blast.cgi). **e** FgSR-GFP localized to the nucleus. Bar = 10 μm

FgSR contains a typical Zn(II)2-Cys6 zinc finger domain and a fungal specific transcription factor domain (Fig. 1d). Although FgSR contains similar domains with Upc2 and Ecm22 (a Upc2 paralog), it shares only 4.26% and 6.32% amino acid identity with Upc2 and Ecm22, respectively. Moreover, FgSR could not complement the sensitivity of yeast Upc2 and Ecm22 mutants to azole compounds (Supplementary Fig. 2a, b), indicating that FgSR is not functionally homologous to Upc2. Confocal microscopic examination revealed that FgSR is localized exclusively in the nucleus (Fig. 1e), which is also different from Upc2. Upc2 is translocated from the cytosol into the nucleus allowing for transcription activation of ergosterol-related genes under the ergosterol-deprived conditions[14,28]. These results indicate that

FgSR serves as a novel transcription factor responsible for ergosterol biosynthesis in *F. graminearum*.

In addition, to confirm that the phenotypic changes observed in the mutant resulted from deletion of *FgSR*, the mutant was complemented with a wild-type FgSR-GFP fusion construct under the control of its native promoter (Supplementary Fig. 3). The defects of ΔFgSR in mycelial growth, sensitivity to azole compounds and ergosterol accumulation were complemented by genetic complementation with the wild-type FgSR-GFP (Fig. 1a–c).

**FgSR binds to the promoters of *FgCYP51* genes**. To check whether ergosterol biosynthesis is regulated by FgSR, we

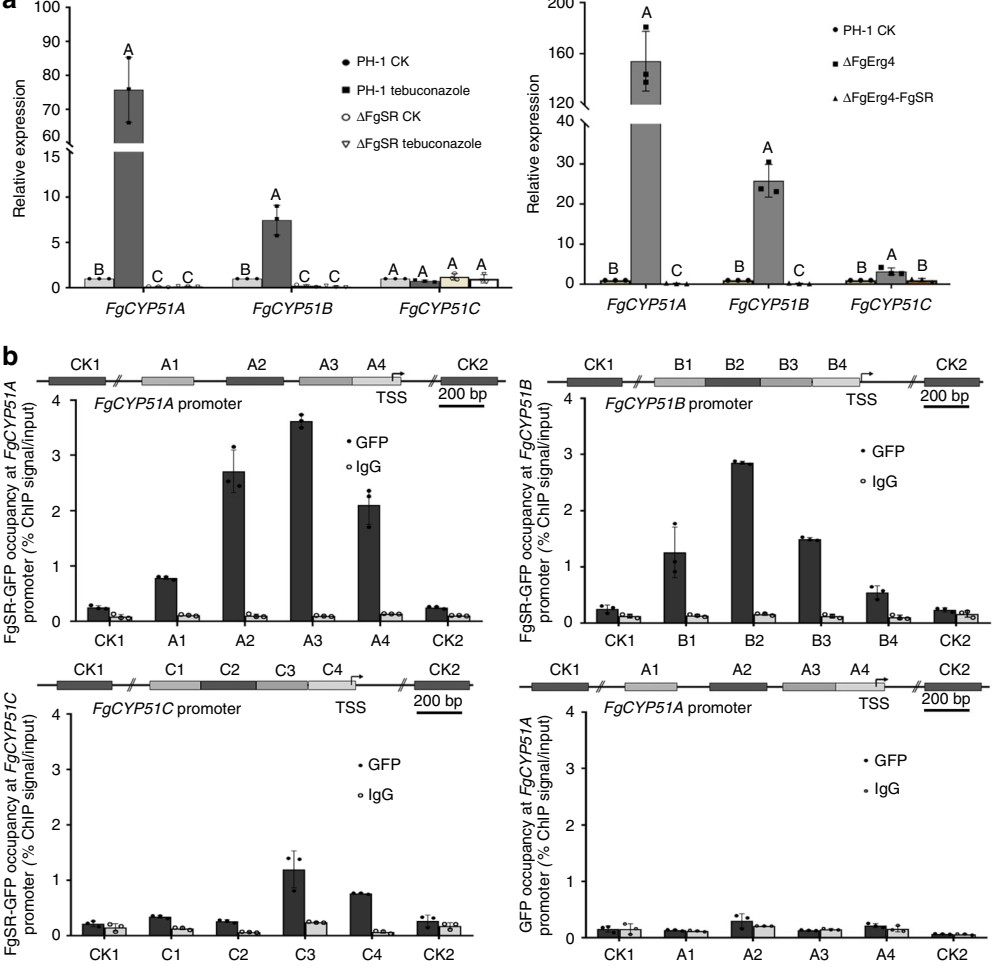

**Fig. 2** FgSR enriches at the promoters of three *FgCYP51* genes and regulates their transcription in *F. graminearum*. **a** The transcription of *FgCYP51A* and *FgCYP51B* was regulated by FgSR, whereas, expression of *FgCYP51C* was modulated by FgSR only under the ergosterol-absent condition. Each strain was cultured in YEPD for 36 h, and then treated with 2.5 μg ml$^{-1}$ tebuconazole for 6 h (left panel). ΔFgErg4 was used to mimic the ergosterol-absent condition (right panel). The expression level of each *FgCYP51* in PH-1 without treatment was referred to 1 and the *FgACTIN* gene was used as the internal control for normalization. Data presented are the mean ± s.d. (*n* = 3). Bars followed by the same letter are not significantly different according to a LSD test at *P* = 0.01. **b** FgSR binds to the promoters of three *FgCYP51* genes, but the enrichment of FgSR at *FgCYP51A* and *FgCYP51B* is more than that at *FgCYP51C*. Each strain was incubated in YEPD liquid medium for 36 h. ChIP- and input-DNA samples were quantified by quantitative PCR assays with primers amplifying different regions indicated on the diagram of each *FgCYP51* promoter. A control reaction was processed in parallel with rabbit IgG and PH-1 transformed only with GFP used as a negative control

examined the transcription of 14-α-demethylase *FgCYP51* genes that encode a key enzyme in sterol biosynthesis[17]. *F. graminearum* contains three paralogous *CYP51* genes, *FgCYP51A, -B*, and *-C*. Quantitative reverse transcription PCR (qRT-PCR) assays showed that expression of *FgCYP51A* and *FgCYP51B*, but not *FgCYP51C* was down-regulated significantly in ΔFgSR under both azole treatment and non-treatment conditions (Fig. 2a). Furthermore, the transcription of *FgCYP51A* and *-B* was greatly induced in the wild type but not in ΔFgSR after treatment with the azole compound tebuconazole (Fig. 2a), but not with iprodione and fludioxonil, compounds with other modes of action (Supplementary Fig. 4). In addition, we tested the transcription of *FgCYP51* genes under the ergosterol-depleting conditions. Our previous study showed that the *FgERG4* deletion mutant (ΔFgErg4) completely loses the ability to synthesize ergosterol since Erg4 catalyses the conversion of ergosta-5,7,22,24-tetraenol to ergosterol in the final step of ergosterol biosynthesis[29]. In this study, ΔFgErg4 was therefore used to mimic the ergosterol-absent condition. As shown in Fig. 2a, the transcription levels of *CYP51A* and *-B* were elevated dramatically in ΔFgErg4 but not in the

double mutant ΔFgErg4-FgSR. It is worthy to note that the transcription level of *CYP51C* was also significantly up-regulated in ΔFgErg4, but not in the double mutant ΔFgErg4-FgSR (Fig. 2a). These results indicate that FgSR regulates the expression of all three *FgCYP51* genes under the ergosterol-depleting conditions.

To further verify that FgSR regulates transcription of *FgCYP51* genes, we determined the binding ability of FgSR to the promoters of *FgCYP51* genes in vivo using chromatin immunoprecipitation and quantitative PCR (ChIP-qPCR) assays. FgSR fused with GFP was transferred into ΔFgSR to serve as the positive strain. In addition, a strain transformed with GFP alone was used as a negative control. ChIP-qPCR analyses showed that FgSR was able to bind to the promoters of *FgCYP51A*, *-B* and *-C* although the enrichment of FgSR at the promoter of *FgCYP51C* was less obvious than that at the promoter of *FgCYP51A* or *FgCYP51B* (Fig. 2b). The low enrichment at the *FgCYP51C* promoter is consistent with the observation that *FgCYP51C* transcription was regulated by FgSR only under ergosterol-absent conditions (Fig. 1c; Fig. 2a). As a control, the GFP enrichment at

*FgCYP51A* promoter in the negative control strain (Fig. 2b) was undetectable, confirming that enrichment determination of FgSR at the promoters of *CYP51A* is reliable. Taken together, these results indicate that FgSR binds to the promoters of *FgCYP51* genes and regulates the transcription of *FgCYP51* genes in *F. graminearum*.

**FgSR is phosphorylated by FgHog1 and interacts with SWI/ SNF.** Under tebuconazole treatment or the lack of *FgERG4*, the transcription of *FgCYP51A* and *-B* was induced dramatically (Fig. 2a; Supplementary Fig. 5a), but the localization, transcription, and protein accumulation of FgSR were not altered under these conditions (Fig. 1d; Supplementary Fig. 5a–c). Moreover, the enrichment of FgSR at the promoter of *FgCYP51A* was not induced after treatment with tebuconazole for 1, 2, 4, and 6 h or by deletion of *FgERG4* (Supplementary Fig. 5d). The result suggests that other components may be required for regulating transcription of *FgCYP51A* and *-B* by FgSR under the ergosterol-deprived condition. To test this hypothesis, we screened a *F. graminearum* cDNA library with the yeast two-hybrid (Y2H) approach, and identified 67 potential FgSR-interacting proteins (Supplementary data 1). One interacting protein was the FgSsk2 MAPKK kinase, and the *FgSSK2* mutant showed increased sensitivity to tebuconazole (Fig. 3a). Furthermore, the association between FgSR and FgSsk2 was confirmed by the Y2H and co-immunoprecipitation (Co-IP) assays (Supplementary Fig. 6a, b), but the direct interaction between FgSR and FgSsk2 was not confirmed by BiFC assay (Supplementary Fig. 6c). Given that FgSsk2 usually functions via the MAPK HOG cascade FgSsk2-FgPbs2-FgHog1 in *F. graminearum*[30,31], we therefore further determined the sensitivity of FgPbs2 and FgHog1 mutants to tebuconazole, and found that ΔFgPbs2 and ΔFgHog1 showed increased sensitivity to tebuconazole similar with ΔFgSsk2 (Fig. 3a). Further, deletion of *FgSSK2*, *FgPBS2* or *FgHOG1* dramatically decreased the expression level of *FgCYP51A* under tebuconazole treatment (Fig. 3b), although the deletion did not affect the enrichment of FgSR at the promoter of *FgCYP51A* (Supplementary Fig. 6d). To explore the relationship between the HOG pathway and tebuconazole senstivity, we found that tebuconazole treatment increased the phosphorylation level and nuclear localization of FgHog1 (Fig. 3c, d). Moreover, phos-tag assays revealed that the amount of the dephosphorylated isoforms of FgSR in ΔFgSsk2, ΔFgPbs2, and ΔFgHog1 was significantly higher than that in the wild type under tebuconazole treatment (Fig. 3e), indicating that tebuconazole treatment activated FgSR via the HOG pathway-mediated phosphorylation. Subsequently, we found that FgSR interacted with the MAP kinase FgHog1 in the Co-IP and BiFC assays, and this interaction was dependent on FgHog1 phophorylation mediated by FgSsk2 and enhanced by tebuconazole treatment (Fig. 3f, g). Importantly, FgSR directly interacted with FgHog1 in the pull down assay, and was phosphorylated by FgHog1 in vitro (Fig. 3h, i). Taken together, these results indicate that tebuconazole treatment activates the HOG pathway, and the activated FgHog1 further phosphorylates FgSR to regulate *FgCYP51A* expression. In addition, in the budding yeast, Ssk2 interacts with Pbs2 and Pbs2 interacts with Hog1 both in vivo and in vitro by the Co-IP and GST-pull down assays[32,33]. In *F. graminerum*, FgSR interacts directly with FgHog1 and the interaction of FgSR and FgSsk2 is likely indirect, probably via the HOG pathway.

To identify the potential phosphorylated sites of FgSR, we performed in silico analysis and found five predicted amino sites which may be phosphorylated by FgHog1 using NetPhos 3.1 Server (http://www.cbs.dtu.dk/services/NetPhos/) (Fig. 3j; Supplementary Fig. 7a). In addition, two independent LC-MS/MS analyses

validated two phosphorylation sites of FgSR in the ΔFgSR::FgSR-GFP strain treated by tebuconazole (Fig. 3j; Supplementary Fig. 7b). We subsequently constructed two strains: one containing a phospho-inactive FgSR and another containing a phosphomimetic FgSR (Fig. 3j). Briefly, the five predicted Ser/Thr phosphorylation residues of FgSR were replaced by alanine to mimic the dephosphorylation condition and by aspartic acid to mimic the phosphorylated condition, respectively. The mutated FgSR$^{T82A/S92A/S102A/T217A/S305A}$ and FgSR$^{T82D/S92D/S102D/T217D/S305D}$ were transformed into ΔFgSR to generate the strain ΔFgSR-C$^{5A}$ and FgSR-C$^{5D}$, respectively. Similar to ΔFgSR and ΔFgHog1, ΔFgSR-C$^{5A}$ exhibited hypersensitivity to tebuconazole (Fig. 3a) and totally disrupted the induced expression of *FgCYP51A* under tebuconazole treatment (Fig. 3b), although these mutations did not affect the enrichment of FgSR at the promoter of *FgCYP51A* (Supplementary Fig. 6d). Moreover, FgSR$^{5A}$ could not be phosphorylated by FgHog1 in vitro (Fig. 3i). In contrast, FgSR-C$^{5D}$ shows similar phenotypes with the wild type under tebuconazole treatment (Fig. 3a, b; Supplementary Fig. 6d), indicating that phosphorylation of FgSR is required for activating *FgCYP51A* gene expression. We further explore the function of each phosphorylation site, and found that the mutants carrying the individual phospho-inactive site did not display increased sensitivity to tebuconazole (Supplementary Fig. 7c), indicating that the five phosphorylation sites of FgSR may share redundant roles. Collectively, these results indicate that FgHog1 phosphorylates FgSR and phosphorylation of FgSR is required for regulating expression of *FgCYP51A*.

The SWI/SNF complex utilizes the energy of ATP hydrolysis to mobilize nucleosomes and remodel chromatin, which regulates transcription of genes located at nucleosome-bound regions[34]. Based on the cDNA library screening assay, we found that FgSR interacted with FgSwp73 that is a homolog of *S. cerevisiae* Swp73. Swp73 as a key subunit of the SWI/SNF complex is essential for the SWI/SNF complex to activate gene transcription[35]. The interaction of FgSR and FgSwp73 was confirmed by the Y2H approach and Co-IP assays (Fig. 4a, b). Moreover, the interaction of FgSR and FgSwp73 was enhanced under tebuconazole treatment (Fig. 4b). To explore whether FgSwp73 is involved the transcriptional regulation of *FgCYP51A* under the treatment with tebuconazole, we first detected the enrichment of FgSwp73, and found that enrichment of FgSwp73 at the *FgCYP51A* promoter could be induced by tebuconazole in the wild type but not in ΔFgSR (Fig. 4c). Second, we deleted another key subunit of SWI/SNF complex FgArp9 which is also responsible for transcriptional regulation of SWI/SNF[36], since we were unable to delete FgSwp73 after obtaining more than 300 ectopic transformants from five independent transformation experiments. As shown in Fig. 4d, e, ΔFgArp9 showed increased sensitivity to tebuconazole and the expression of *FgCYP51A* in ΔFgArp9 was significantly lower than that in the wild type after tebuconazole treatment. Third, the enrichment of FgArp9 at the *FgCYP51A* promoter also relied on FgSR (Fig. 4c). Fourth, the enrichment of histone H3 at the promoter of *FgCYP51A* significantly elevated in ΔFgSR and ΔFgArp9, indicating that nucleosome occupancy at the *FgCYP51A* promoter is controlled by SWI/SNF and is associated with *FgCYP51A* transcription (Fig. 4f). These results indicate that the SWI/SNF complex interacts with FgSR during the transcriptional regulation of *FgCYP51A*.

Interestingly, the mutated FgSR$^{T82A/S92A/S102A/T217A/S305A}$ could not interact with FgSwp73, indicating that the phosphorylation of FgSR is required for the interaction of FgSR and FgSwp73 (Fig. 4a, b). Further, we found that in ΔFgSsk2, ΔFgHog1, and ΔFgSR-C$^{5A}$, the enrichments of FgSwp73 and FgArp9 at the *FgCYP51A* promoter were reduced and the

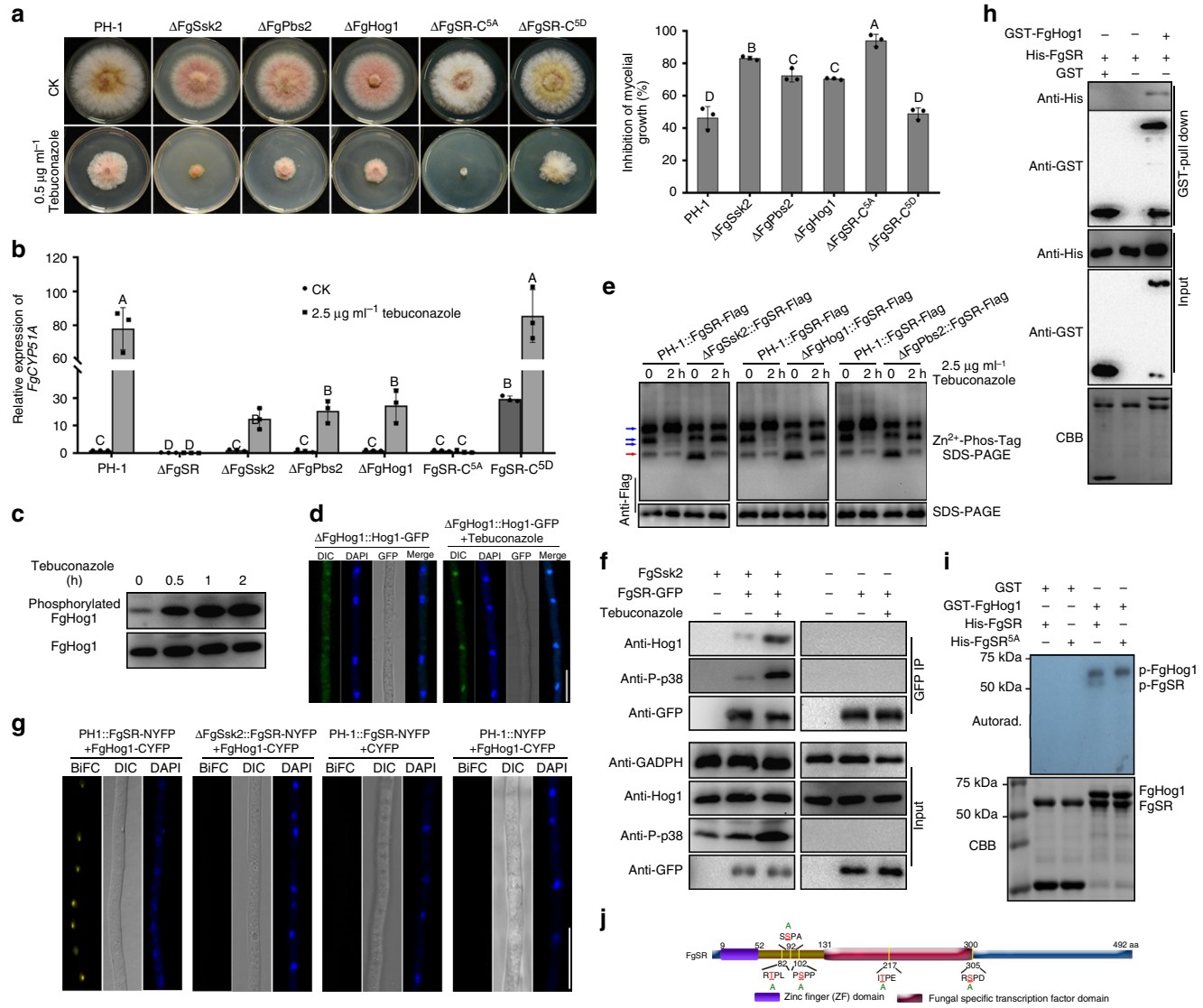

**Fig. 3** Phosphorylation of FgSR mediated by FgHog1 regulates the expression of *FgCYP51A*. **a** Deletion of kinase in the HOG cascade or mutations at the predicted phosphorylation sites within FgSR led to increased sensitivity to tebuconazole. **b** Comparisons of expression level of *FgCYP51A* among the above strains after the treatment with 2.5 μg ml⁻¹ tebuconazole for 6 h. The expression level of *FgCYP51A* in the wild type without treatment was referred to 1. Data presented are the mean ± s.d. (*n* = 3). Bars followed by the same letter are not significantly different according to a LSD test at *P* = 0.01. **c** The phosphorylation of FgHog1 was increased by the treatment with 2.5 μg ml⁻¹ tebuconazole. PH-1 was treated with tebuconazole for 0, 0.5, 1 and 2 h after incubated in YEPD for 36 h. **d** Localization of FgHog1-GFP with 2.5 μg ml⁻¹ tebuconazole treatment for 0 and 2 h. Nuclei were stained with 4′,6-diamidino-2-phenylindole (DAPI). DIC, differential interference contrast. Bar = 10 μm. **e** Loss of FgSsk2, FgPbs2 or FgHog1 decreased the phosphorylation of FgSR with or without tebuconazole treatment. The dephosphorylated and phosphorylated FgSR are indicated with red and blue arrows, respectively. **f** FgHog1 interacts with FgSR in the co-immunoprecipitation (Co-IP) assay and the interaction is dependent on FgSsk2 and enhanced by tebuconazole treatment. The strains were treated with (+) or without (−) tebuconazole for 2 h after incubated in YEPD for 36 h. **g** FgHog1 interacts with FgSR in the nucleus under the treatment with 2.5 μg ml⁻¹ tebuconazole, and the interaction is dependent on the FgSsk2 by bimolecular fluorescence complementation (BiFC) assays. Bar = 10 μm. **h** The interaction of FgHog1 and FgSR assayed using GST pull-downs. Proteins were also detected by staining with Coomassie brilliant blue (CBB). **i** Phosphorylation of FgSR by Hog1 in vitro. Phosphorylated proteins were resolved by SDS-PAGE and detected by autoradiography (upper panel). GST-FgHog1 and His-FgSR were also detected by staining with CBB (lower panel). **j** Position of the five predicted phosphorylated amino acid residues in FgSR

enrichment of histone H3 at the promoter of *FgCYP51A* is increased (Fig. 4c, f). These results suggest that the phosphorylation of FgSR is required for the interaction with the SWI/SNF complex.

**FgSR forms a homodimer and binds to the target gene promoters**. In the cDNA library screening assay, we found that FgSR interacts with itself (Supplementary data 1), indicating that FgSR may form a homodimer. We further confirmed FgSR homodimerization by using the Y2H, Co-IP, and BiFC assays (Fig. 5a–c). Subsequently, we tried to identify the functional domain in FgSR that is required for homodimerization. As shown in Fig. 5a, the Y2H assays with a series of truncated FgSRs revealed that the middle linker (ML) domain between zinc finger and fungal specific transcription factor domains is essential for homodimerization of FgSR. BiFC and Co-IP assays also confirmed the requirement of ML domain for the formation of FgSR

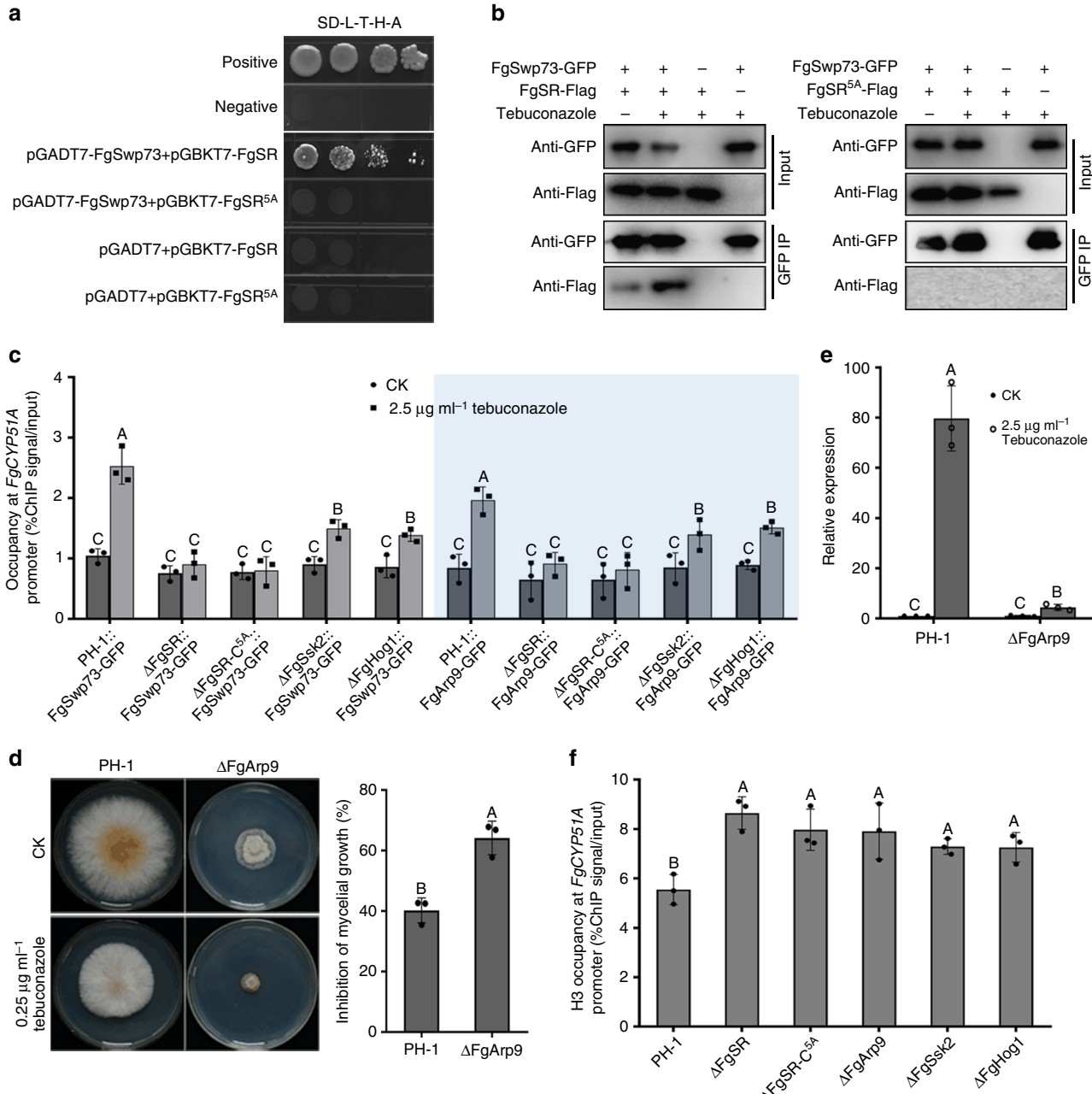

**Fig. 4** Phosphorylated FgSR interacts with the SWI/SNF complex to regulate transcription of *FgCYP51A*. **a** The yeast two-hybrid (Y2H) assay revealed that the 5 phosphorylation residues of FgSR are critical for its interaction with FgSwp73. **b** FgSR interacts with FgSwp73 in the co-immunoprecipitation (Co-IP) assay. The relative strains were treated with or without tebuconazole for 2 h after incubated in YEPD for 36 h. **c** The enrichment of FgSwp73-GFP and FgArp9-GFP at the *FgCYP51A* promoter among the wild-type PH-1, ΔFgSR, ΔFgSsk2, ΔFgHog1, and ΔFgSR-C$^{5A}$. ChIP- and input-DNA samples were quantified by quantitative PCR assays with the primer pair A2 indicated in Fig. 2b, and rabbit IgG was used as a control. **d** Deletion of the SWI/SNF complex subunit *FgARP9* led to increased sensitivity to tebuconazole. A 5-mm mycelial plug of PH-1, ΔFgArp9 was inoculated on PDA alone or supplemented with 0.25 µg ml$^{-1}$ tebuconazole (left panel), and the mycelial growth inhibition was calculated for each strain (right panel). **e** Comparisons of the expression of *FgCYP51A* between PH-1 and ΔFgArp9 after the treatment with 2.5 µg ml$^{-1}$ tebuconazole for 6 h. The expression of *FgCYP51A* in the wild type without treatment was referred to 1. **f** The enrichment of histone H3 at the *FgCYP51A* promoter among the above strains. ChIP- and input-DNA samples were quantified by quantitative PCR assays with the primer pair A2 indicated in Fig. 2b, and rabbit IgG was used as a control. Data presented are the mean ± s.d. ($n = 3$). Bars followed by the same letter are not significantly different according to a LSD test at $P = 0.01$

homodimer (Fig. 5b, c). Phenotypic characterization of the strain lacking the ML domain (ΔFgSR-C$^{ΔML}$) showed that, similar to ΔFgSR and the strain lacking the zinc finger (ZF) domain (ΔFgSR-C$^{ΔZF}$), ΔFgSR-C$^{ΔML}$ displayed the defects in response to azole compounds, ergosterol biosynthesis, induction of *FgCYP51A* expression by tebuconazole treatment (Fig. 5d–f),

although FgSR$^{ΔML}$-GFP localizes in nucleus (Fig. 5g). ChIP-qPCR assay showed that similar to ΔFgSR-C$^{ΔZF}$, FgSR$^{ΔML}$ could not bind to the promoter of *FgCYP51A* (Fig. 5h). Previous studies have found that zinc finger transcription factor usually forms dimer to bind target sequence and the domain which is required for dimerization is primarily located in the C-terminus next to the

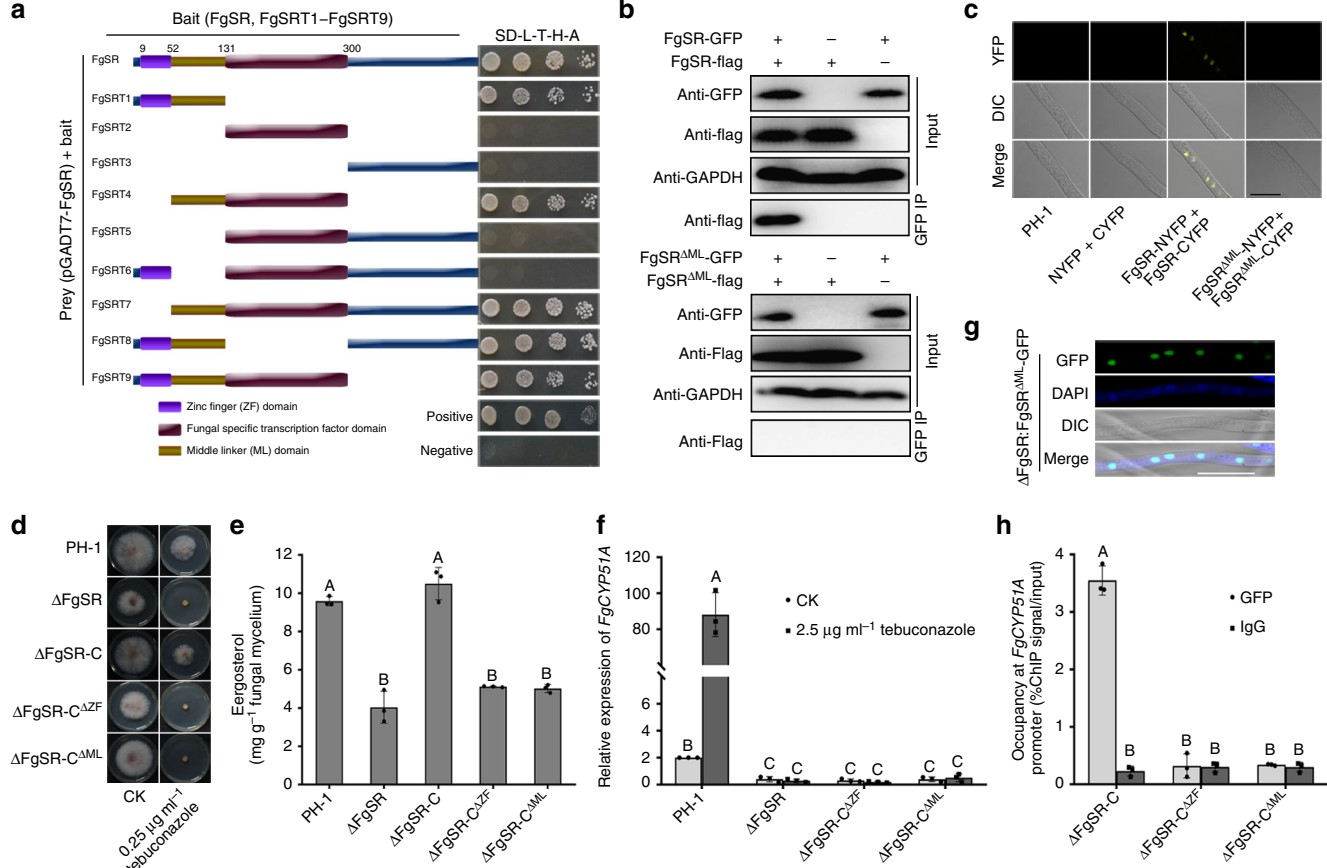

**Fig. 5** The homodimerization of FgSR is required for its binding to the *FgCYP51A* promoter. **a** The yeast two-hybrid (Y2H) assay revealed that the middle linker (ML) domain of FgSR is required for its homodimerization. Serial dilutions of yeast cells (cells ml$^{-1}$) transferred with the prey (pGADT7-FgSR) and bait (a series of truncated FgSR (indicated in the left panel) constructs were assayed for growth on SD-Leu-Trp-His plates (right panel). **b** FgSR lacking of the ML domain did not interact with itself in the co-immunoprecipitation (Co-IP) assay. **c** FgSR lacking of the ML domain did not interact with itself in the bimolecular fluorescence complementation (BiFC) assay. FgSR-NYFP/FgSR$^{\Delta ML}$-NYFP, and FgSR-CYFP/FgSR$^{\Delta ML}$-CYFP were co-transformed into the wild type. YFP signals were observed using confocal microscopy. Bar = 10 μm. **d** The strain lacking the zinc finger (ZF) domain (ΔFgSR-C$^{\Delta ZF}$) and ML domain (ΔFgSR-C$^{\Delta ML}$) as well as the ΔFgSR mutant displayed greatly increased sensitivity to tebuconazole. **e** The strains ΔFgSR-C$^{\Delta ZF}$ and ΔFgSR-C$^{\Delta ML}$ exhibited reduced ergosterol content similar to ΔFgSR. **f** In ΔFgSR-C$^{\Delta ZF}$ and ΔFgSR-C$^{\Delta ML}$, the expression of *FgCYP51A* was significantly decreased and could not be induced after the treatment with 2.5 μg ml$^{-1}$ tebuconazole for 6 h. **g** FgSR$^{\Delta ML}$-GFP is still localized in the nucleus. Bar = 10 μm. **h** ΔFgSR-C$^{\Delta ZF}$ and FgSR$^{\Delta ML}$-GFP did not enrich at the *FgCYP51A* promoter. ChIP- and input-DNA samples were quantified by quantitative PCR assays with the primer pair A2 indicated in Fig. 2b, and rabbit IgG was used as a control. Data presented are the mean ± s.d. (*n* = 3). Bars followed by the same letter are not significantly different according to a LSD test at *P* = 0.01

DNA binding domain[37]. These results indicate that the homodimerization of FgSR mediated by the ML domain is required for its binding to the target gene promoter.

**FgSR targets multiple genes involved in ergosterol biosynthesis.** To evaluate whether other ergosterol biosynthesis genes are also regulated by FgSR, we performed a genome-wide ChIP-Seq assay in two independent experiments, and found that FgSR bound to the 1-kb promoter region upstream of 119 open reading frames (ORFs) (Supplementary Fig. 8a; Supplementary data 2). The Kyoto Encyclopedia of Genes and Genomes (KEGG) pathway analysis showed that these 119 genes are mainly involved in steroid and terpenoid backbone biosynthesis, metabolism of tryptophan, glyoxylate and dicarboxylate, the phagesome, and valine, leucine and isoleucine degradation (Supplementary Fig. 8b). In addition, we conducted ChIP-qPCR assays with three selected genes, *FgCTK1* (FGSG_13888), *FgERG6A* (FGSG_05740), and *FgERG6B* (FGSG_02783), to verify ChIP-Seq data (Supplementary Fig. 8c). Consistently, qRT-PCR assays showed that the expression levels of these genes were significantly down-regulated in ΔFgSR (Supplementary Fig. 8d).

Among the 119 genes bound by FgSR, 20 are involved in ergosterol biosynthesis (Supplementary data 2). ChIP-qPCR assays verified that FgSR was able to bind to the promoter of each of these genes (Fig. 6a). Moreover, the RNA-Seq assay showed that 14 out of these 20 genes were down-regulated in ΔFgSR (Fig. 6a; Supplementary data 3). Consistent with this observation, ΔFgSR displayed increased sensitivity to multiple SBIs, which target different sterol biosynthetic enzymes (Fig. 6a, b). Overall, these results strongly indicated that FgSR is a master transcriptional factor in the ergosterol biosynthesis pathway.

**Identification of the *cis*-element of FgSR.** To identify the FgSR-binding *cis*-element in the target promoters, we analyzed five ergosterol-related genes (*FgCYP51A*, *FgCYP51B*, *FgERG6A*, *FgERG6B*, and *FgCTK1*) using the multiple EM for motif elicitation (MEME) program. These five genes were down-regulated in RNA-Seq and bound by FgSR in ChIP-Seq data. The MEME

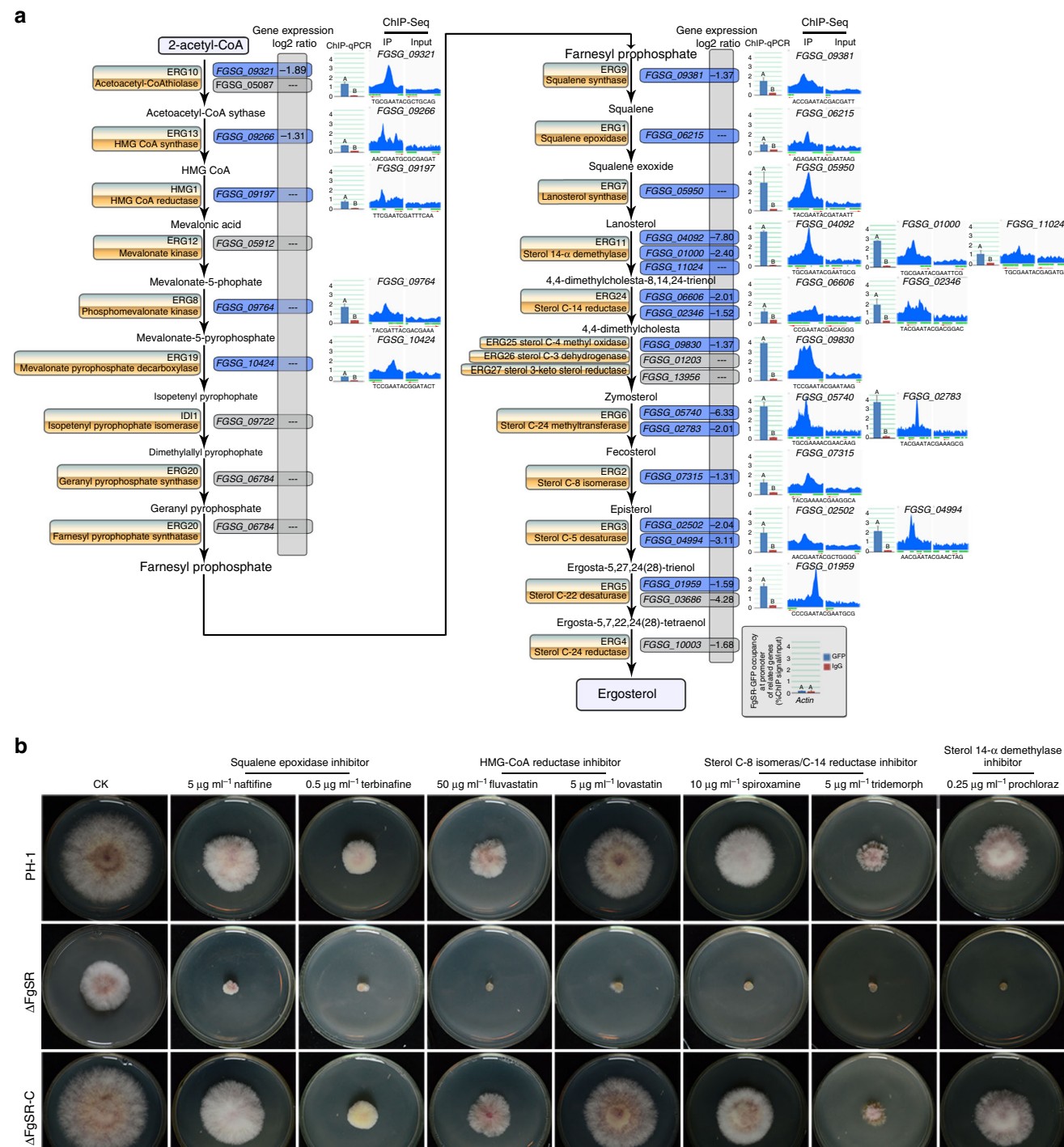

**Fig. 6** FgSR is a master regulator of ergosterol biosynthesis in *F. graminearum*. **a** ChIP-Seq and -qPCR assays showed that FgSR binds to the promoters of 20 ergosterol biosynthesis genes indicated in blue. RNA-Seq further confirmed that 14 out of the above 20 genes were down-regulated in the ΔFgSR mutant. *FgACTIN* was use as a negative control in the ChIP-qPCR assay. ChIP- and input-DNA samples were quantified by quantitative PCR assays with the primer pair A2 indicated in Fig. 2b, and rabbit IgG was used as a control. The red arrow indicates the gene direction from 5′ to 3′. The green bar represents the gene promoter region, which is covered during ChIP-Sequencing. "black dashed line" indicates no change in gene transcription determined by the RNA-Seq assay (Supplementary data 3). The sequences below the kurtosis graphs represent the *cis*-elements within the promoters of the 20 ergosterol pathway genes identified by MEME analysis. Values on the bars followed by the same letter are not significantly different at $P = 0.01$. **b** FgSR regulates *F. graminearum* sensitivity to sterol biosynthesis inhibitors (SBIs) targeted to different biosynthetic enzymes. A 5-mm mycelial plug of each strain was inoculated on PDA supplemented with 5 μg ml⁻¹ naftifine, 0.5 μg ml⁻¹ terbinafine, 5 μg ml⁻¹ fluvastatin, 5 μg ml⁻¹ lovastatin, 10 μg ml⁻¹ spiroxamine, 5 μg ml⁻¹ tridemorph, or 0.25 μg ml⁻¹ prochloraz and then incubated at 25 °C for 3 days. As a control, each strain was cultured on PDA without supplementation

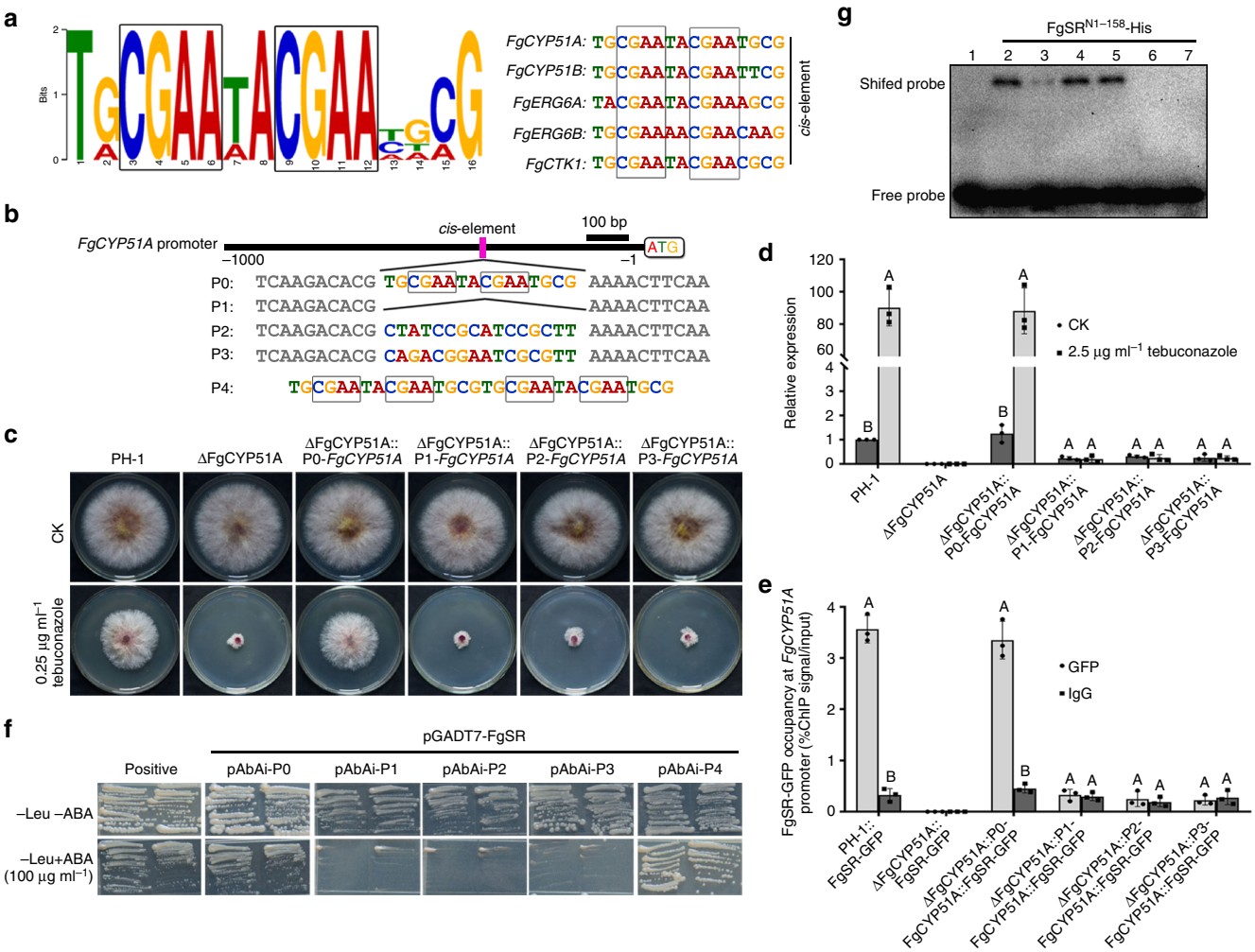

**Fig. 7** Identification of *cis*-element of FgSR in *F. graminearum*. **a** The putative sequence of the *cis*-element obtained by analyzing the promoters of five target genes with the MEME program. Two CGAA-repeated sequences within the *cis*-elementare indicated by the black squares. **b** Schematic representation of modified *cis*-element in the promoter of *FgCYP51A*. Two CGAA-repeated sequences are indicated by the black squares. **c** The *cis*-element-deleted (ΔFgCYP51A::P1-FgCYP51A) or -mutated (ΔFgCYP51A::P2-FgCYP51A and ΔFgCYP51A::P3-FgCYP51A) strains displayed increased sensitivity to tebuconazole. **d** In the strains ΔFgCYP51A::P1-FgCYP51A, ΔFgCYP51A::P2-FgCYP51A, and ΔFgCYP51A::P3-FgCYP51A, the expression of *FgCYP51A* was reduced and could not be induced by the treatment with 2.5 μg ml⁻¹ tebuconazole. The expression level of *FgCYP51A* in the wild type without treatment was set to 1. **e** ChIP-qPCR assay revealed that FgSR could not bind to the *FgCYP51A* promoter lacking the *cis*-element or containing a mutated *cis*-element. ChIP- and input-DNA samples were quantified by quantitative PCR assays with the primer pair A2 indicated in Fig. 2b; rabbit IgG was used as a control. Data presented are the mean ± s.d. (*n* = 3). Bars followed by the same letter are not significantly different according to a LSD test at *P* = 0.01. **f** FgSR binds to the *cis*-element in the FgCYP51A promoter as indicated by yeast one-hybrid (Y1H) assays. The native FgCYP51A promoter (pAbAi-P0), *cis*-element-deleted and mutated FgCYP51A promoters (pAbAi-P1 to -P3) or the two repeats of the *cis*-element (pAbAi-P4) was used as a bait, and the pGADT7-FgSR as the prey. **g** Verification of the binding of FgSR with the *cis*-element by electrophoretic mobility shift assay (EMSA). Lane 1 and 2, biotin-labeled motif of P0 (as indicated in Fig. 7b) without FgSR^N1−158 (lane 1) or with FgSR^N1−158 (lane 2); lane 3–5, biotin-labeled motif of P0 with 100 fold excess of unlabelled motif of P0 (lane 3), unlabeled mutated motif of P2 (lane 4), unlabeled mutated motif of P3 (lane 5) with FgSR^N1−158; lanes 6 and 7, biotin-labeled mutated motif of P2 and P3 respectively with FgSR^N1−158

analysis identified a predicted 16-bp *cis*-element containing two CGAA-repeated sequences (indicated by black squares in Fig. 7a), which is consistent with that provided by the ChIP-Seq analysis (Supplementary Fig. 8e). Further analysis revealed that this *cis*-element existed in the promoters of 64 (54%) FgSR-binding genes identified from the ChIP-Seq assay including 20 sterol biosynthesis genes (Fig. 6a; Supplementary data 2).

To determine whether this *cis*-element is essential for transcription of *FgCYP51A* regulated by FgSR in vivo, we constructed *cis*-element-deleted (ΔFgCYP51::P1-FgCYP51) or -mutated (ΔFgCYP51::P2-FgCYP51 and ΔFgCYP51::P3-FgCYP51) strains using an in situ complementation strategy (Fig. 7b). Briefly, we knocked out the *FgCYP51A* open reading frame and its

promoter using HPH-HSV-tk as a marker, and then complemented the mutant with full length *FgCYP51A* under control of its native promoter as well as with its promoter *cis*-element deleted or mutated as shown in Fig. 7b. Sensitivity determination showed that the *cis*-element-deleted or -mutated strains did not complement the sensitivity of ΔFgCYP51 to tebuconazole (Fig. 7c). Moreover, the transcription levels of *FgCYP51* were decreased dramatically in these strains treated with or without tebuconazole (Fig. 7d). Furthermore, ChIP-qPCR assays revealed that FgSR did not bind to the *FgCYP51A* promoter lacking the *cis*-element or containing mutated *cis*-elements (Fig. 7e). Further, the yeast one-hybrid (Y1H) approach was used to further verify that this *cis*-element is responsible for the binding of FgSR to the promoter of *FgCYP51A*

in vitro. Briefly, the full-length cDNA of FgSR was fused into the GAL4 activation domain in the vector pGADT7. The 1000-bp *FgCYP51A* promoter containing the P0 (the wild-type *cis*-element), or the mutated P1, P2, or P3 *cis*-elements were cloned into the pAbAi vector (Fig. 7b). Y1H assays showed that FgSR was able to directly interact with the 1-kb *FgCYP51A* promoter region carrying the wild-type *cis*-element (P0), but did not bind to the promoter containing a deleted or mutated *cis*-element P1, P2 or P3 (Fig. 7f). Moreover, FgSR bound to P4, a sequence that contained only the two repeats of the *cis*-element, confirming that the binding of FgSR with the *cis*-element was independent of other sequences in the *FgCYP51A* gene promoter. In addition, we obtained the FgSR$^{N1-158}$ (containing the ZF and ML domains)-His recombinant protein. Consistently, EMSA confirmed that FgSR$^{N1-158}$ binds the identified motif (Fig. 7g). Taken together, these results confirmed that this *cis*-element is required for the binding of FgSR to promoters of sterol biosynthesis genes.

Given that FgSR is different from SREBP and Upc2 orthologs in protein structure or the regulatory mechanism for sterol biosynthesis, we searched FgSR orthologs in 27 eukaryotic species belonging to five subphylums and ranging from the fungus *S. cerevisiae* to the mammal *H. sapiens*. As shown in Fig. 8a, FgSR and its orthologs were conserved in the *Sordariomycetes* and *Leotiomycetes*, which are phylogenetically closely related *Ascomycota* based on analysis of the conserved RNA polymerase II subunit RPB2. Moreover, MEME analysis of 41 *CYP51* (also named as *ERG11*) orthologs from 27 eukaryotic species uncovered that the FgSR-binding *cis*-element only existed in *Sordariomycetes* and *Leotiomycetes* fungi, indicating an ancestral relationship for FgSR homologs and their corresponding *cis*-elements (Fig. 8a; Supplementary Table 1). Further, we knocked out FgSR orthologs in the *Sordariomycetes* fungus *Fusarium oxysporum* and the *Leotiomycetes* fungus *Botrytis cinerea*, and found that the mutants displayed increased susceptibility to SBIs but not to other antifungal compounds (Fig. 8b; Supplementary Fig. 9). These results indicate that FgSR homologs are a novel class of transcription factors, and they regulate ergosterol biosynthesis in *Sordariomycetes* and *Leotiomycetes* fungi.

**FgSR is indispensable for virulence in *F. graminearum*.** Given that *F. graminearum* is an important pathogen on various cereal crops[21], we therefore were interested in evaluating virulence of ΔFgSR on flowering wheat heads and corn silks. As shown in Fig. 9a, scab symptoms caused by ΔFgSR were restricted to the inoculated spikelets fifteen days after inoculation. On the other hand, the wild type and the FgSR-complemented strains caused serious scab symptoms on wheat heads under the same condition. Similar results were observed in corn silk infection assays. In comparison with extensive necrosis in the wild type, ΔFgSR only caused limited necrosis and discoloration near the inoculation sites four days after inoculation (Fig. 9b). These results indicate that FgSR is required for full *F. graminearum* pathogenicity.

Our study found that deletion of FgSR reduced ~50% growth, but caused a dramatic decrease in virulence (Fig. 9a, b). We were therefore interested in exploring the mechanism of reduced virulence in ΔFgSR. DON, as an important virulence factor, plays crucial roles in the spread of *F. graminearum* within host plants[18,38]. Determination of DON accumulation showed that DON accumulation on sterilized wheat kernels inoculated with ΔFgSR was 85% lower than the wild type at 20 days after inoculation (Fig. 9c). Likewise, the transcription of DON biosynthesis genes *FgTRI1*, *FgTRI6*, and *FgTRI8*, as shown by qRT-PCR, was decreased in ΔFgSR in culture under the trichothecene biosynthesis inducing (TBI) conditions[39] (Fig. 9d). These results indicate that the decreased virulence of ΔFgSR may

at least partially result from the fact that the mutant produced less DON.

Two-benzoxazolinone (BOA) is a major phytoalexin produced by wheat during defense response to various biotic stresses[40]. Previous studies found that the degradation of the benzoxazolinone class of phytoalexins is important for virulence of *Fusarium*[41]. Interestingly, we found that ΔFgSR was hypersensitive to BOA (Fig. 9e). To explore the response to BOA regulated by FgSR, we knocked out six genes that may be responsible for detoxification of BOA in *F. graminearum*[41], and found that *FgFDB1*, *−2*, and *−3* mutants showed increased sensitivity to BOA (Fig. 9e), which indicate that FgFdb1/2/3 play important roles in BOA detoxification. Further, qRT-PCR assays revealed that the transcription of *FgFDB1*, *−2*, and *−3* was significantly reduced in ΔFgSR (Fig. 9f). In addition, the *FgFDB1*, *−2*, and *−3* mutants displayed decreased virulence on wheat head (Fig. 9a, b). These results indicated that the defect in scavenging the phytolexin contributes to reduced virulence of the ΔFgSR mutant.

During infection by plant pathogenic fungi, DNA damage stress triggered by plant defense responses such as increased levels of reactive oxygen species or reactive nitrogen species, pose a threat to successful infection of pathogens[42,43]. Thus, we determined the sensitivity of ΔFgSR to the DNA damage agents 4-nitroquinoline 1-oxide (4-NQO) and hydroxyurea (Hu), and found that this mutant displayed greatly increased sensitivity to these compounds (Fig. 9g). Further, we selected five genes involved in DNA replication based on the ChIP-Seq data (Supplementary data 2), and confirmed enrichment of the FgSR *cis*-element at the promoters of these five genes (Fig. 9h). Moreover, qRT-PCR assays showed that these five genes could be up-regulated by 4-NQO treatment in the wild type but not in ΔFgSR (Fig. 9i). These results indicated that FgSR may control virulence at least in part by regulating the response to DNA damage stress during infection.

## Discussion

In this study, we identified a novel sterol biosynthesis regulator FgSR in *F. graminearum*. FgSR is structurally different from the SREBP and Upc2 orthologs used to regulate sterol biosynthesis in other fungi. Moreover, both the SREBP and Upc2 homologs are retained in *F. graminearum*, but no longer function in regulating sterol biosynthesis. Importantly, FgSR employs a unique regulatory mechanism, different from SREBP and Upc2, in modulating sterol biosynthesis in *F. graminearum*. Consistently, the FgSR binding *cis*-element DNA sequence is also distinct from those of Upc2 and SREBP[44,45] (Fig. 8a). Phylogenetic analysis showed that FgSR orthologs are conserved only in *Sordariomycetes* and *Leotiomycetes* fungi. As experimental support of conserved function, deletion of FgSR homolog in *F. oxysporum* (a *Sordariomycetes* fungus) and in *B. cinerea* (a *Leotiomycetes* fungus) led to increased sensitivity to SBIs. Interestingly, the binding *cis*-element for FgSR also exists in the promoters of relevant genes only in *Sordariomycetes* and *Leotiomycetes* fungi, indicating that FgSR orthologs and their cognate *cis*-elements arose together in a common evolutionary ancestor. These results indicate that FgSR homologs functionally displace the Upc2 and SREBP and serve as a novel class of sterol biosynthesis regulator in *Sordariomycetes* and *Leotiomycetes*.

The regulatory mechanism of sterol biosynthesis mediated by the transcription factor SREBP in most fungi and metazoa, or Upc2 in *Saccharomycotina*, has been well characterized[13,46]. In the current study, we found that FgSR possesses a novel mechanism for regulating sterol biosynthesis (Fig. 10), which differs from SREBP and Upc2 orthologs. First, FgSR has a stable presence in the nucleus, which is independent of treatment with

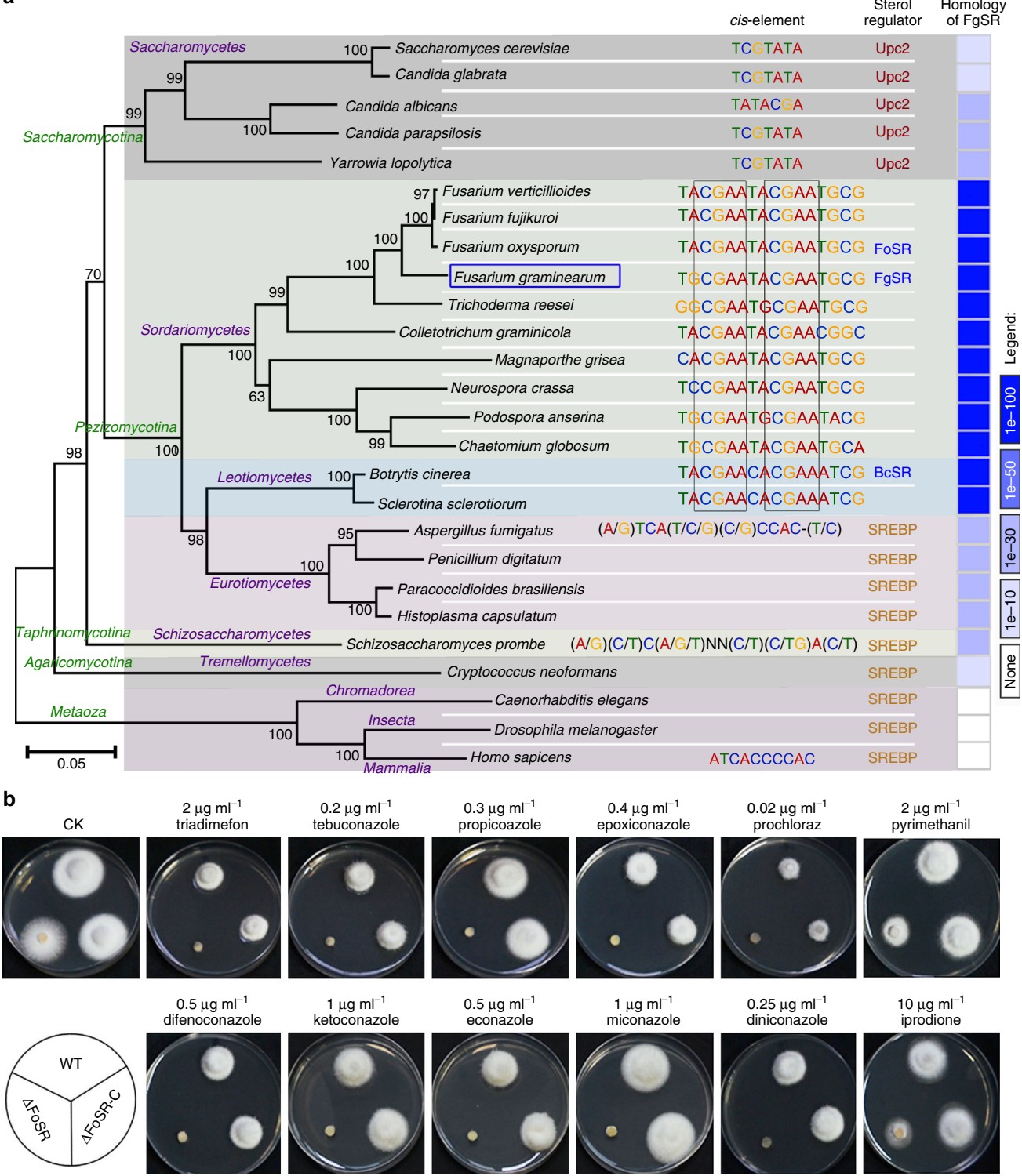

**Fig. 8** FgSR homologs and their binding *cis*-element are highly conserved in *Sordariomycetes* and *Leotiomycetes* fungi. **a** FgSR orthologs share high homology in *Sordariomycetes* and *Leotiomycetes* fungi, and the FgSR-binding *cis*-elements within the 1.2-kb promoters of the *CYP51* orthologous genes were also detected only in these fungi. The phylogenetic tree was constructed based on the amino acid sequences of RPB2 from 27 eukaryotic species with Mega 5.0 using the neighbor-joining method. The bootstrap values from 1000 replications are indicated on the branches. The homology among FgSR orthologs from 27 species were analyzed using the BLASTMatrix tool on the Comparative Fungal Genomics Platform (http://cfgp.riceblast.snu.ac.kr/). The depth of blue color on the right indicates the level of homology with FgSR. **b** Deletion of the FgSR ortholog in the *Sordariomycetes* fungus *Fusarium oxysporum* led to elevated sensitivity to sterol biosynthesis inhibitor (SBIs). A 5-mm mycelial plug of each strain was inoculated on PDA alone or supplemented with a SBI compound, and then incubated at 25 °C for 2 days. The non-sterol biosynthesis inhibitors, iprodione and pyrimethanil, were used as a control

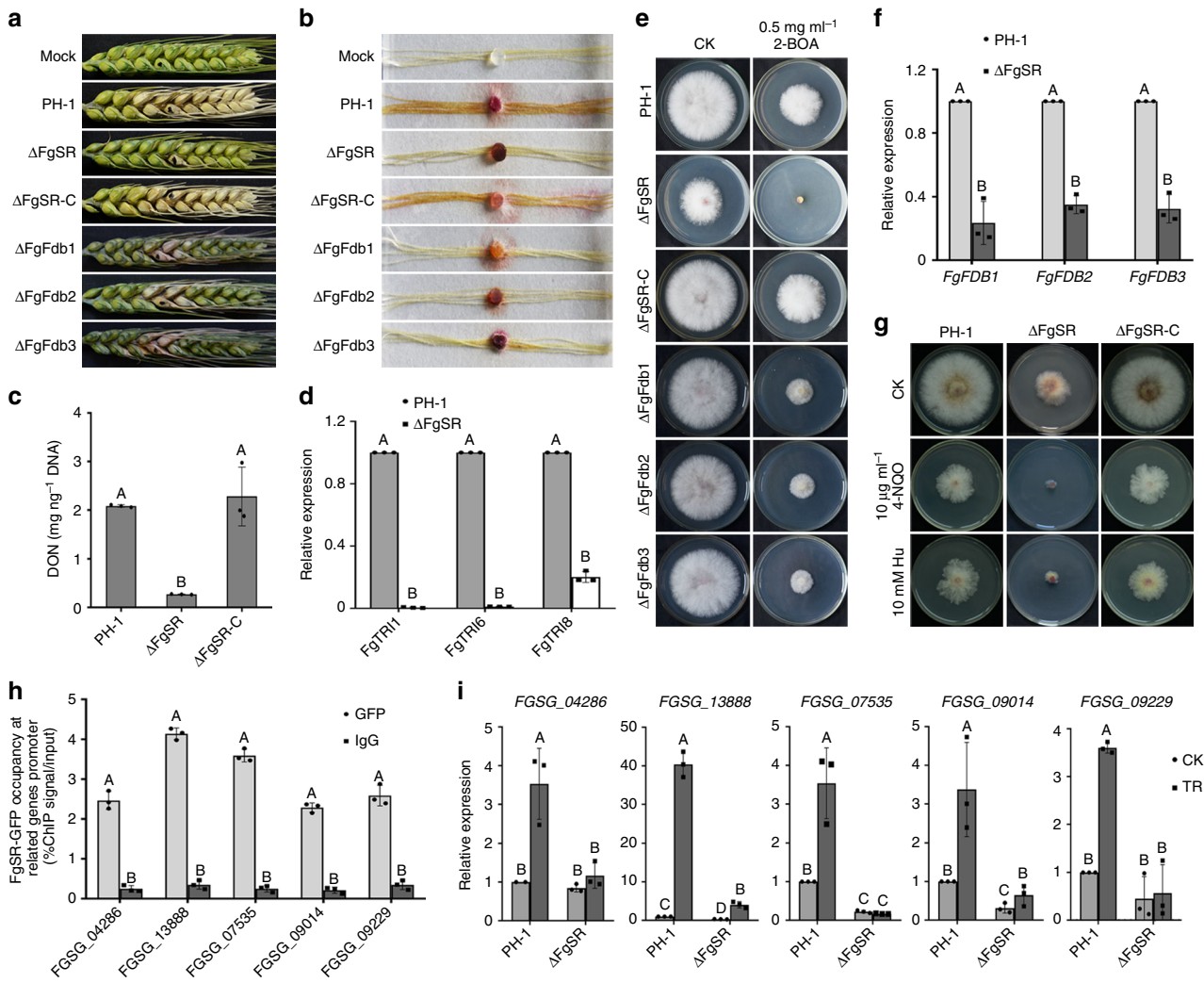

**Fig. 9** FgSR is required for virulence in *F. graminearum*. **a** Flowering wheat heads were point inoculated with a conidial suspension at 10^5 conidia ml^−1 of each strain and infected wheat heads were photographed 15 days after inoculation (dai). **b** Corn silks were inoculated with mycelial plugs of each strain and examined 4 dai. **c** The amounts of DON (per ng DNA) produced by the wild type, ΔFgSR and ΔFgSR-C in infected wheat kernels was determined after 20 days of inoculation. **d** Relative expression levels of DON biosynthetic gene *FgTRI1*, *FgTRI6*, and *FgTRI8* in wild type and the ΔFgSR mutant grown in TBI for two days. **e** Sensitivity of PH-1, ΔFgSR, ΔFgSR-C, ΔFgFdb1, −2 and −3 to the phytoalexin 2-benzoxazolinone (BOA). **f** *FgFDB1*, *FgFDB2* and *FgFDB3* were induced in the wild type but not in the ΔFgSR mutant after the treatment with 1 mg ml^−1 BOA. The expression level of each gene in the wild type with treatment was set to 1. **g** Sensitivity of PH-1, ΔFgSR and ΔFgSR-C to DNA damage agent4-nitroquinoline 1-oxide (4-NQO) and hydroxyurea (Hu). **h** The enrichment of FgSR-GFP at the promoters of five genes related to DNA replication. ChIP- and input-DNA samples were quantified by PCR using primers in each gene promoter. Rabbit IgG was used in as a control. **i** Relative expression levels of DNA replication-related genes in PH-1 and the ΔFgSR mutant after the treatment with 50 µg ml^−1 4-NQO for 4 h. The expression level of each gene in the wild type without treatment was normalized to 1. Data presented are the mean ± s.d. (*n* = 3). Bars followed by the same letter are not significantly different according to a LSD test at *P* = 0.01

an azole compound or ergosterol starvation. Conversely, the homologous SREBP proteins from *H. sapiens*[47], *A. fumigatus*[11], *C. neoformans*[48], and *S. pombe*[9], are transferred into nucleus only after their N terminus are released from membrane attachment under conditions of low intracellular sterol. Similarly in *S. cerevisiae*, Upc2 is trafficked into the nucleus following treatment with sterol-depleting drugs and disassociation of its LBD with ergosterol[14]. Second, cellular FgSR protein levels are not altered by treatment with tebuconazole or by depleting the concentration of ergosterol. In contrast, the transcription and translation of Upc2 is up-regulated under the sterol-deprived condition created by the treatment of lovastatin, that is a competitive inhibitor of 3-hydroxy-3-methylglutaryl-coenzyme A (HMG-CoA) reductase and mimics ergosterol starvation in cells[12]. Third, FgSR protein levels bound at the promoters of target genes do not increase

under ergosterol-deprived conditions. Unlike FgSR, Upc2 protein levels at target gene promoters dramatically increased after the treatment with lovastatin in *S. cerevisiae*[12]. Fourth, phosphorylation of FgSR mediated by the HOG pathway kinase regulates its activity and its interaction with the SWI/SNF complex. However, post-translational modification is not reported for transcriptional activation of SREBP and Upc homologs. The transcriptional activation of SREBP and Upc2 homologs both rely on the change of localization from cytosol to nucleus[6,14]. Finally, in *S. pombe*, *S. cerevisiae*, *C. neoformans*, and *C. albcans*, Hog1 homologs suppress the transcription of ergosterol biosynthesis genes, subsequently the Hog1 mutants showed reduced sensitivity to the azole compounds[49–51]. In *F. graminearum*, however, we find that the kinases of HOG pathway positively regulates the transcription of ergosterol biosynthesis genes. Taken together,

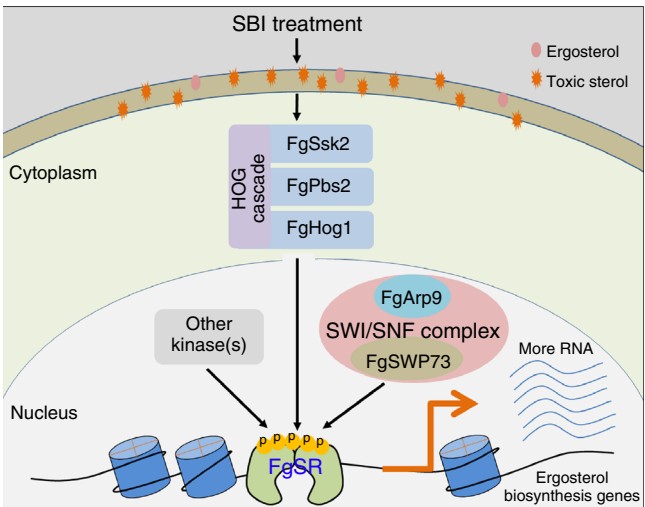

**Fig. 10** A proposed model for the regulation of sterol biosynthesis mediated by FgSR in *F. graminearum*. Under the treatment with sterol biosynthesis inhibitors (SBIs), the HOG cascade FgSsk2-FgPbs2-FgHog1 is activated in *F. graminearum*. Subsequently, FgSR is strongly phosphorylated by activated FgHog1 and other kinase(s). The highly phosphorylated FgSR recruits the SWI/SNF complex to remodel chromatin, and subsequently induces the high transcription levels of sterol biosynthesis genes

transcriptional activation of FgSR in *F. graminearum* is dependent on phosphorylation, which is different from what is known for SPEBP and Upc2.

In eukaryotes, phosphorylation of transcription factors mediated by various kinases have been found to regulate their DNA binding ability and protein accumulation[52]. In mammalian systems, phosphorylation of the runt-related transcription factor (RUNX3) at the threonine 173 (T173) residue disrupts its DNA binding activity[53]. In *Arabidopsis thaliana*, phosphorylation of *Arabidopsis* seed shattering 4-related 3 (Asr3) by MEK4 (MAMP-activated MPK4), and phosphorylation of trihelix transcription factor (GT-1) by calmodulin kinase II both increase their DNA binding activity at the target gene promoters[52,54]. In fission yeast, phosphorylation of SpSre1 (a SREBP homolog) by Hhp2, a member of the casein kinase family, reduces its protein quantity by accelerating its degradation[55]. However, in this study, we discovered that phosphorylation of FgSR mediated by the MAP kinase FgHog1 did not alter protein accumulation or DNA binding ability of FgSR to its target promoters under ergosterol-depleted conditions. Additionally, we found that the strain ΔFgSR-C[5A] was more sensitive to tebuconazole than ΔFgHog1, the transcription of *FgCYP51* was still slightly induced in ΔFgHog1 under the tebuconazole treatment, and partial phosphorylated FgSR presented in ΔFgHog1 (Fig. 3a, b, e), which indicates that other kinase(s) besides the predominant FgHog1 may also be involved in phosphorylation of FgSR.

Surprisingly, we found that the phosphorylated FgSR interacts with the SWI/SNF complex to regulate target gene transcription. In mammalian, activator protein 1 (AP1), as a cooperative transcription factor of glucocorticoid receptor (GR) binds the regulatory elements prior to environmental stimuli and subsequently potentiate chromatin accessibility and GR binding[56,57]. But the detailed mechanism of regulating chromatin accessibility is unknown. In this study, we report that the phosphorylated FgSR interacts with the SWI/SNF complex to regulate the transcription of sterol biosynthesis genes upon tebuconazole treatment. Although there are several reports about that transcription factor or histone demethylase may recruit the chromatin remodelling complex or histone modification complex to regulate gene

expression in *S. cerevisiae*, rice, and *Arabidopsis*[58–61], the functions of protein post translational modification in this processing remains unclear.

Previous studies have shown that the adaptation to hypoxia is important for fungal virulence[11,62], and sterol biosynthesis is associated with the hypoxic growth during fungal infection[1,48]. In *A. fumigatus, C. neoformans,* and *C. albicans*, deletion of transcription factor SREBP or Upc2 causes reduced virulence, which mainly results from decreased growth under hypoxia or the abolishment of hypoxia-induced filamentation[48,63]. However, deletion of FgSR did not affect vegetative growth in hypoxia although it caused decreased biosynthesis of aurofusarin (Supplementary Fig. 10) which is not required for full virulence in *F. graminearum*[64]. These results suggest that FgSR regulation of virulence may employ a strategy that is different from what is known for Upc2 and SREBP. In this study, we firstly found that FgSR regulates DON production that is important for disease spread[38]. Farnesyl pyrophosphate (FPP) as the immediate precursor of DON biosynthesis is derived from the isoprenoid biosynthetic pathway, which is also the first step of sterol biosynthesis[2]. Moreover, five isoprenoid biosynthesis genes, which are targeted by FgSR are significantly down-regulated in ΔFgSR (Supplementary Fig. 11). Thus, deletion of FgSR results in decreased FPP biosynthesis, which subsequently leads to reduced DON production. In addition, FgSR also modulates the response to phytoalexin BOA and DNA damage agent 4-NQO. Studies in *Fusarium pseudograminearum* and *C. albicans* have proved that the ability of pathogens to cope with phytoalexins and DNA damage agents plays important roles in their virulence[41,43]. Therefore, FgSR controls virulence by regulating DON biosynthesis and response to environmental stresses in *F. graminearum*.

In a conclusion, we have identified a novel transcription factor FgSR and its *cis*-element regulating ergosterol biosynthesis in *F. graminearum*. The ML domain within FgSR controls its DNA binding ability. Furthermore, the phosphorylation of FgSR regulates its transcriptional activity and its interaction with the SWI/SNF complex. Interestingly, phylogenetic and genetic analyses showed that FgSR homologs may serve as a main sterol biosynthesis regulator in *Sordariomycetes* and *Leotiomycetes* fungi. In addition, FgSR also functions in virulence by modulating DON biosynthesis, and fungal responses to phytoalexins and DNA damage.

## Methods

**Strains and phenotype determination**. The wild-type strain PH-1 (NRRL 31084) of *F. graminearum* was used as a parental strain for transformation experiments. Mycelial growth of the wild-type and the resulting transformants were assayed on PDA, CM or MM, and the image were taken three days after inoculation[65]. To determine sensitivity to various stresses, 5-mm mycelial plugs of each strain taken from a 3-day-old colony edge were inoculated on PDA supplemented without/with each stress agent, or under normoxia/hypoxia, then incubated at 25 °C for 2 or 3 days in the dark. The concentrations for each compound were indicated in figure legends. Three biological replicates were used for each strain and each experiment was repeated three times independently.

To determine ergosterol content, mycelia were grown in yeast extract peptone dextrose (YEPD) liquid medium for 36 h. The powdered mycelium was combined with 8 ml 3 M KOH in ethanol and incubated at 80 °C on a shaker in a water bath for 1 h. Extraction mixtures were centrifuged for 15 min at 4000 rpm and the supernatant diluted with 2 ml distilled water. Ergosterol was extracted from the supernatant by two successive applications of 5 ml hexane. The organic fractions were combined and evaporated under nitrogen, then re-suspended in 500 μl of methanol and analyzed for ergosterol content using an Agilent 1100 HPLC system (Palo Alto, CA, USA).

To determine DON production, healthy wheat kernels were soaked in sterile water for 18 h, and then laid on a sieve for 2 h to remove excess water. A 50-g sample of healthy wheat kernels was then sterilized and inoculated with five mycelial plugs of each strain. A negative control was prepared with the wheat kernels inoculated with five agar plugs without mycelium. After incubation at 25 °C for 20 days, 1.5 g portions of ground grain were extracted with 10 ml of

acetonitrile-water (86/14, v/v) and shaken for overnight. Extracts were purified with a PuriToxSRDON column TC-T200 (TrilogyAnalytical Laboratory, Washington, MO, USA), and then evaporated under a stream of nitrogen at 40 °C. Finally, the extracts were resolubilized in 2 ml of methanol/water (1/1, v/v) for HPLC analysis using an Agilent 1100 HPLC system (Palo Alto, CA, USA).The amount of *F. graminearum* DNA in each sample was used as a reference.

For virulence on wheat spikelets, conidia harvested from 4-day-old cultures of *F. graminearum* strains were resuspended in 0.01% (vol/vol) Tween 20 and adjusted to $10^5$ conidia $ml^{-1}$. Ten microliters aliquot of a conidial suspension was injected into a floret in the middle spikelet of a wheat head of susceptible cultivar Jimai22 at early anthesis. There were 20 replicates for each strain. For virulence on corn silks, a 5-mm mycelial plug of each strain was inoculated on the middle of corn silks, and then cultured for 4 days. Ten replicates were used for each strain. Each experiment was repeated four times.

### Mutant generation, complementation and in situ displacement of the *cis*-element.

The gene deletion mutants of *F. graminearum* were constructed using a PEG mediated protoplast transformation method[66]. To get *F. graminearum* protoplasts, the fresh mycelium were treated with driselase (D9515, Sigma, Sigma, St. Louis, MO), lysozyme (RM1027, RYON, Shanghai, China) and cellulose (RM1030, RYON, Shanghai, China). The primers used to amplify the flanking sequences for each gene are listed in Supplementary data 4. Putative gene deletion mutants were identified by PCR assays, and the *FgSR* deletion mutant was further confirmed by a Southern blot assay (Supplementary Fig. 3).

A FgSR-GFP fusion fragment, obtained using the double-joint PCR approach, was co-transformed with XhoI-digested pYF11 into the yeast strain XK1–25 using the Alkali-CationTM Yeast Transformation Kit (MP Biomedicals, Solon, USA) to generate a FgSR-GFP fusion vector[65]. Using a similar strategy, FgSR lacking ML domain (FgSR$^{ΔML}$)-GFP, FgSR containing five mutated phosphorylation sites (FgSR$^{5A}$)-GFP and FgHog1-GFP fusion cassettes were constructed. Complementation of ΔFgSR with FgSR-, FgSR$^{ΔML}$-, or FgSR$^{5A}$-GFP vector was acheived using geneticin (*NEO*) as the second selectable marker.

For in situ displacement of the motif in the *FgCYP51A* promoter, a gene replacement cassette carrying the hygromycin (HPH) resistance gene, the herpes simplex virus thymidine kinase gene (HSV) and the trpC promoter, was constructed by double-joint PCR[67]. Firstly, the flanking sequences of *FgCYP51A* gene motif were amplified and then were fused with a fragment containing HPH and HSV to construct a recombinant DNA cassette. This cassette was used to transform protoplasts of *F. graminearum* strain PH-1 to generate the native promoter-FgCYP51A deletion mutant (ΔFgCYP51). After identification using the primers in Supplementary data 4, ΔFgCYP51 was complemented with the full-length native promoter-FgCYP51A (P0-FgCYP51) fragment as well as with the *FgCYP51A* and its *cis*-element-deleted (P1-FgCYP51) or -mutated (P2- or P3-FgCYP51A) fragments using floxuridine as the selected marker. The P0-, P1-, P2- and P3-FgCYP51A fragments were sequenced before transformation.

### Chromatin immunoprecipitation (ChIP)-Seq and ChIP-qPCR analyses.

ChIP was performed according to described protocol[68] with additional modifications. Briefly, fresh mycelia were cross-linked with 1% formaldehyde for 15 min and then stopped with 125 mM glycine. The cultures were ground with liquid nitrogen and re-suspended in the lysis buffer (250 mM, HEPES pH 7.5, 150 mM NaCl, 1 mM EDTA, 1% Triton, 0.1% DeoxyCholate, 10 mM DTT) and protease inhibitor (Sangon Co., Shanghai, China). The DNA was sheared into ~300 bp fragments with 20 pulses of 10 s and 20 s of resting at 35% amplitude (Qsonica*sonicator, Q125, Branson, USA). After centrifugation, the supernatant was diluted with 10 × ChIP dilution buffer (1.1% Triton X-100, 1.2 mM EDTA, 16.7 mM Tris–HCl, pH 8.0 and 167 mM NaCl). Immunoprecipitation was performed using monoclonal anti-GFP ab290 (Abcam, Cambridge, UK; 1:500 dilution) antibody or anti H3 (ab1791, Abcam, Cambridge, UK; 1:500 dilution) antibody together with the protein A agarose (sc-2001, Santa Cruz, CA, USA) respectively. After washed orderly by low salt wash buffer (one time; 150 mM NaCl, 20 mM Tris-HCl (PH 8.0), 0.2% SDS, 0.5% TritonX-100, 2 mM EDTA), high salt wash buffer (one time; 500 mM NaCl, 20 mM Tris-HCl (PH 8.0), 0.2% SDS, 0.5% TritonX-100, 2 mM EDTA), LiCl wash buffer, TE buffer (two times; 100 mM Tris-HCl (PH 8.0), 10 mM EDTA), DNA was eluted from beads with elution buffer (1% SDS, 0.1 M NaHCO$_3$) and then immunoprecipitated by ethanol after washing, eluting, reversing the cross-linking and digesting with proteinase K.

Further, ChIP-enriched DNA was sent for sequencing on an Illumina HiSeq 2500 (Genergy Bio, Shanghai, China), or used for quantitative PCR analysis using SYBR green I fluorescent dye detection with the relative primers (Supplementary data 4). Relative enrichment values were calculated by dividing the immunoprecipitated DNA by the input DNA. ChIP-Seq was performed twice independently and ChIP-qPCR was independently repeated three times.

### Yeast complementation assay.

Full-length cDNA of FgSR was cloned into the pYES2 vector (Invitrogen Co., CA, USA), and transformed into the corresponding yeast mutant. The wild-type strain BY4741 and the mutant transformed with an empty pYES2 vector were used as controls. For complementation assays, the yeast transformants were grown at 30 °C on YPRG (1% yeast extract, 2% bactopeptone,

2% galactose) medium supplied with an azole compound. The experiments were independently repeated three times.

### Yeast one- and two-hybrid (Y1H and Y2H) assays.

To confirm the binding of FgSR to its *cis*-element, the Y1H assay was carried out using the Matchmaker One-Hybrid Library Construction and Screening Kit (Clontech, http://www.clontech. com/) according to the manufacturer's instruction. The different fragments were cloned into the pAbAi vector. The constructs were transferred into yeast strain Y1HGold and were tested by synthetic medium (SD)-Ura with 100 ng $ml^{-1}$ AbA (Aureobasidin A). The full length of FgSR was amplified from wild-type cDNA by PCR and cloned into the pGADT7 vector, and then was transferred into yeast strain Y1HGold which contained the different pAbAi vectors. Interactions of FgSR with different fragments were tested by SD-Leu with 100 ng $ml^{-1}$ AbA. The interaction of pGADT7-p53 and p53-AbAi plasmids was used as the positive control. Three independent experiments were performed to confirm the Y1H results.

To construct plasmids for Y2H analyses, the coding sequence of each tested gene was amplified from cDNA of the wild type with primer pairs indicated in Supplementary data 4. The cDNA fragment was inserted into the GAL4-binding domain vector pGBKT7 and GAL4-activation domain vector pGADT7 (Clontech, Mountain View, CA, USA) respectively. The pairs of Y2H plasmids were co-transformed into *S. cerevisiae* strain Y2H Gold following the lithium acetate/single-stranded DNA/polyethylene glycol transformation protocol. In addition, the plasmid pair pGBKT7–53 and pGADT7 served as a positive control. The plasmid pair pGBKT7-Lam and pGADT7 was used as a negative control. Transformants were grown at 30 °C for 3 days on SD lacking Leu and Trp, and then serial dilutions of yeast cells (cells $ml^{-1}$) were transferred to SD stripped of His, Leu and Trp and Ade to assess protein–protein interaction. Three independent experiments were performed to confirm the Y2H results.

To search FgSR-interacting proteins, we performed Y2H screens. FgSR was cloned into the yeast vector pGBKT7 (Clontech). A *F. graminearum* cDNA library was constructed in the Y2H vector pGADT7 using total RNA extracted from mycelia and conidia. The Y2H Gold that were co-transformed with cDNA library as well as FgSR-pGBKT7 were directly selected using SD-Trp-Leu-His. Approximately 500 potential yeast transformants containing cDNA clones interacting with FgSR were further confirmed in selection medium SD-Trp-Leu-His-Ade.

### Co-IP assays.

The GFP- and 3 × Flag-fusion constructs were verified by DNA sequencing and transformed in pairs into PH-1. Transformants expressing a pair of fusion constructs were confirmed by western blot analysis. For Co-IP assays, fresh *F. graminearum* mycelia (200 mg) were finely ground and suspended in 1 ml of extraction buffer (50 mM Tris-HCl, pH 7.5, 100 mM NaCl, 5 mM EDTA, 1% Triton X-100, 2 mM phenylmethylsulfonylfluoride (PMSF) containing 10 μl of protease inhibitor cocktail (Sangon Co., Shanghai, China). After homogenization with a vortex shaker, the lysate was centrifuged at 10,000 × g for 20 min at 4 °C and the supernatant was incubated with the anti-GFP (ChromoTek, Martinsried, Germany) agarose. Proteins eluted from agarose or from supernatant were analyzed by western blot detection with monoclonal anti-Flag (A9044, Sigma, St. Louis, MO; 1:10,000 dilution), anti-GFP (ab32146, Abcam, Cambridge, UK; 1:10,000 dilution), anti-Hog1 (#9211, Cell Signaling Technology Inc., Boston, MA, USA; 1:2000 dilution) or anti-P-p38 (sc-165978, Santa Cruz, CA, USA; 1:2000 dilution) antibody. The samples were also detected with monoclonal anti-GAPDH antibody (EM1101, Hangzhou HuaAn Biotechnology co., Ltd.; 1:10,000 dilution) as a reference. After inoculation respectively with secondary antibody anti-rabbit IgG-HRP (sc-2317, Santa Cruz 315 Biotech, Heidelberg, Germany; 1:10,000 dilution) or anti-mouse IgG-HRP (sc-2791, Santa Cruz 315 Biotech, Heidelberg, Germany; 1:10,000 dilution), chemiluminescence was detected. The experiment was conducted independently three times. All blots were imaged by the ImageQuant LAS 4000 mini (GE Healthcare) (Supplementary Fig. 12).

### Bimolecular fluorescence complementation (BiFC) assay.

The FgSR-CYFP, FgSR$^{ΔML}$-CYFP, FgHog1-CYFP, and FgSsk2-CYFP fusion constructs were generated by cloning the related fragments into pHZ68. Similarly, the FgSR-NYFP and FgSR$^{ΔML}$-NYFP fusion constructs were generated by cloning the related fragments into pHZ65. A pair of FgSR-NYFP and FgHog1-CYFP constructs were co-transformed into PH-1 and ΔFgSsk2. Construct pairs of FgSR-NYFP and FgSsk2-CYFP, FgSR-CYFP and FgSR-NYFP, FgSR$^{ΔML}$-CYFP and FgSR$^{ΔML}$-NYFP were co-transformed into protoplasts of PH-1. Construct pairs of FgSR-NYFP and CYFP, FgHog1-CYFP and NYFP, NYFP + CYFP were also co-transformed into PH-1 to serve as negative controls. Transformants resistant to both hygromycin and zeocin were isolated and confirmed by PCR. YFP signals were examined with a Zeiss LSM780 confocal microscope (Gottingen, Niedersachsen, Germany). The experiment was conducted independently three times.

### Phos-tag analysis.

The FgSR-Flag fusion construct was transferred into the wild-type strain PH-1, ΔFgSsk2, ΔFgPbs2, and ΔFgHog1, respectively. Each protein sample was extracted as described in Co-IP assays. The resulting protein samples were resolved on 8% SDS-polyacrylamide gels prepared with 25 μM Phos binding

reagent acrylamide (APE × BIO, F4002) and 100 µM $ZnCl_2$. Gels were electrophoresed at 20 mA/gel for 3–5 h. Prior to protein transfer, gels were first equilibrated three times in transfer buffer containing 5 mM EDTA for 5 min and further equilibrated in transfer buffer without EDTA twice for 5 min. Protein from the $Zn^{2+}$-phos-tag acrylamide gel to the PVDF membrane was transferred for 4–5 h at 100 V at 0 °C, and finally the membrane was analyzed by western blotting using an anti-Flag antibody (A9044, Sigma, St. Louis, MO; 1:10,000 dilution). The experiment was conducted independently three times.

**Electrophoretic mobility shift assay (EMSA).** cDNA encoding the N terminal 1–158 amino acids of FgSR (FgSR$^{N1-158}$) was amplified and cloned into pET32a vector to generate His-tagged protein. The resulting construct was transformed into the *E. coli* strain BL21 (DE3) after verifying the cDNA sequence. The recombinant FgSR$^{N1-158}$-His was purified by Ni sepharose beads and eluted by imdazole. Probe DNAs labeled with or without biotin at the 3 end and their reverse complementary chains were synthesized by Thermofisher Company. For EMSA, we used LightShift™ Chemiluminescent EMSA Kit 20148 (Thermofisher Company, USA). Reaction mixtures containing purified FgSR $^{N1-158}$ and biotin-labeled probe were incubated for 20 min at room temperature. The reactions were electrophoresed on 6% polyacrylamide gel in 0.5 × TBE, and transferred to a positively charged nylon membrane (Millipore, USA). Signals were detected using Chemiluminescent Substrate in the kit according to the manufacturer's instruction. The experiment was conducted independently three times.

**In vitro protein-binding and phosphorylation assays.** The full length His-FgSR and mutated His-FgSR$^{5A}$ were expressed in *E. coli* and purified using Ni sepharose beads by company (Detai Biologics, Nanjing, China). The GST fusion proteins encoding the FgHog1 and FgPbs2$^{EE}$ were expressed in *E. coli* DH5a and purified using glutathione-sepharose beads (GE Healthcare). To test in vitro binding between His- and GST-tagged proteins, 3 µg of GST-tagged protein or GST (negative control) that was still bound to the glutathione beads was mixed with 10 µg of His-tagged protein and rocked for 2 h at 4 °C. The beads were washed six times with cold TBS and resuspended in 5 × SDS sample buffer, and GST pull-down proteins were then analyzed by western blotting using the monoclonal mouse Anti-His6-Peroxidase antibody (11965085001, Sigma, St. Louis, MO; 1:2000 dilution Abcam, Cambridge, MA, USA) and the monoclonal mouse anti-GST antibody (sc-53909, Santa Cruz 315 Biotech, Heidelberg, Germany; 1:2000 dilution). The experiment was conducted two times independently. For in vitro phosphorylation assays, 1-µg sample of FgHog1 was activated with 0.2 µg of FgPbs2$^{EE}$ in the presence of kinase buffer (50 mM Tris-HCl pH 7.5, 10 mM MgCl2, 2 mM DTT) and 50 mM ATP. After 20 min at room temperature, 2 µg of His-FgSR or His-FgSR$^{5A}$ protein was added to the previous mixture together with [γ-$^{32}$P]ATP incubated for 4 h at room temperature. The reaction was terminated by the addition of 4 × SDS loading buffer. Labelled proteins were resolved by SDS-PAGE, stained, dried, and detected by autoradiography. The experiment was conducted two times independently.

**Reporting summary.** Further information on experimental design is available in the Nature Research Reporting Summary linked to this article.

## Data availability
The ChIP-Seq data has been deposited in the NCBI BioProject database with accession code PRJNA520738. Other relevant data supporting the findings of the study are available in this article and its Supplementary Information files.

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

## Acknowledgements

We thank Prof. Jinrong Xu (Purdue University) for providing PH-1::Hog1-GFP strains, and FgPbs2 and FgHog1 mutants. This research was supported by the National Natural Science Fund for Distinguished Young Scholar (31525020), the National Key R&D Program of China (2017YFC1600900), National Science Foundation (31571945, 31871910, 31801675), China Agriculture Research System (CARS-3–1–15), the Fundamental Research Funds for the Central Universities (2017FZA6014), Dabeinong Funds for Discipline Development and Talent Training in Zhejiang University, China Postdoctoral Science Foundation (2017M620250) and The International Postdoctoral Exchange Fellowship Program. Additional support was provided by award 2014–67013–21561 from the USDA National Institute of Food and Agriculture.

## Author contributions

Z.L. conducted most of the experiments. Z.L., Y.Y., and Z.M. designed most of the experiments and directed the project; Y.J. performed chromatin remodeling work; Y.C. performed the analyses of RNA-Seq data; Z.L., Y.Y., Z.M., H.C.K., and P.H. analyzed the data and wrote the paper.

## Additional information

**Competing interests:** The authors declare no competing interests.

