## [Peer Review File · Nature Communications]

Reviewers' comments:

Reviewer #1 (Remarks to the Author):

The manuscript by Liu et al describes the identification and detailed characterisation of a here-to-unrecognised pathway that regulates sterol biosynthesis. After providing evidence that *Fusarium* homologues of known sterol regulatory pathways from other organisms did not have this role in *F. graminearum*, the authors undertook an elegant loss-of-function genetic screen to find transcription factors that contribute to basal tolerance to sterol biosynthesis inhibitors. The authors went on to characterise how the protein functions and the signalling and chromatin remodelling complexes involved in its regulation of sterol biosynthesis.

Although the bulk of the paper (and title) focuses on the role of the identified transcription factor and signalling pathways in sterol biosynthesis regulation the work provides tantalising evidence that the transcription factor has a broader role in regulation of transcription associated with the pathogenesis process. For this reason the title could be made slightly broader.

I have relatively minor comments

Line 136: "restore the sensitivity of yeast *Upc2* and *Ecm22* mutants to azole compounds" could be interpreted that the *Upc2* and *Ecm22* mutants are actually resistant to these compounds which I suspect is not the intended meaning. Perhaps "restore" should be replaced with "complement". Likewise in the supplementary figure 2 legend "restore sensitivity" implies the mutants are actually resistant.

Line 200: The link between the Y2H result and the *Ssk2* mutant result does not read very well. Perhaps "Among them, we observed that the mutant of *Fgssk2* also showed increased sensitivity to..." could be changed to "One interacting protein was the *FgSsk2* MAPKK kinase and mutants in the encoding gene showed increase sensitivity to..."

Line 206: In Figure 3e it appears the difference between expression of *Cyp51A* under tebuconazole treatment is only slightly reduced in the *ssk2* mutant compared to Ph1. Should this sentence on line 206 read "to 89.2%"

The analysis of deoxynivalenol production in the *FgSR* mutant seemed an obvious course of investigation given the in plant a phenotype of the mutant not spreading from the inoculation point in wheat and the shared upstream pathways for sterol and deoxynivalenol biosynthesis which were shown to be regulated by *FgSR* including via direct binding to the promoters. Likewise the results of the analysis of DNA damage stress also seemed logical for inclusion given the ChIP-seq data supported enrichment of *FgSR* at the promoters of some of the genes involved in the response to these stresses. It is less clear why the sensitivity to BOA was tested and how *FgSR* regulates the expression of the detoxification genes given these were not identified in the ChIP-seq data. Whilst the data are interesting around BOA a better justification of why these genes were investigated and an explanation of how *FgSR* might regulate their expression should be included.

Figure 2: promoter is spelled incorrectly in the vertical axes of part b. It might also be useful to align the order of the different regions assayed in the promoters with the bars as they are currently in opposite orientation in the promoter diagram to the graphs below.

In figure 2a, *delta-erg4* induced ergosterol deficiency clearly induces *cyp51a* expression. However, in Supp Fig 5, *cyp51a* does not seem to be induced in the *delta-erg4::FgSR-GFP* strain which should functionally equivalent to the *delta-erg4* strain used in figure 2a. Can the reason for this discrepancy be explained?

Line 408: vegetative not vegetable

Overall the work is a very convincing characterisation of the novel sterol regulation system in *Fusarium* and related ascomycete fungi. I suspect this paper will be quite influential in the field as the findings are highly novel. The statistical analyses have been performed with quantitative data but the tests used to identify groups that differ (defined on bar charts by letters) have not been explained in the figure legends or the methods. Having said that the data is still very convincing so this is just a technicality.

Donald Gardiner

Reviewer #2 (Remarks to the Author):

Fusarium graminearum is an important plant pathogenic fungus. In this study, the authors showed that six orthologs of known regulators of ergosterol biosynthesis were dispensable. The FgSR mutant was then found to have increased sensitivity to tebuconazole. Enrichment of FgSR at the promoter of Cyp51A/B was observed. The authors then screened yeast two-hybrid library and identified FgSsk2 and Swp73 as two of the FgSR-interacting proteins. Five putative phosphorylation sites were identified and characterized in FgSR. Mutations at these five sites affected the interaction of FgSR with Swp73. The authors then showed that the SNF complex may be recruited to the Cyp51A promoter via FgSR. They also found that FgSR was enriched in other genes related to ergosterol synthesis by ChIP-seq. Putative FgSR binding sites were identified. Interestingly, FgSR proteins were only found in Sordariomycetes and Leotiomycetes. Based on these data, the authors proposed that FgSR recruits the chromatin remodeling factors to regulate fungal sterol biosynthesis.

To this reviewer, some of the conclusions were not well-supported by data presented in this manuscript.

Major concerns:

1. The subcellular localization and enrichment of FgSR on the CYP51 promoters were not affected by tebuconazole treatment. One major experimental evidence to show that phosphorylation of FgSR recruited the SNF complex to these promoters is that 5A mutation affected the interaction between FgSR and Swp73. Because this result is critical to the conclusions of this manuscript, it is important to prove the interaction between FgSR and Swp73 and the effect of 5A mutations on their interactions by co-IP assays and BiFC assays. A simple yeast two-hybrid assay is not sufficient.

Although FgSsk2 was dispensable for tebuconazole induced enrichment, Swp73 and Arp9 were. Is that somewhat contradictory? According to the descriptions, phosphorylation of FgSR by FgSsk2 and other kinase is important for recruiting the SNF complex to the FgSR bound promoters.

2. I have serious doubt about the interaction of FgSsk2 with FgSR and the direct role of FgSsk2 in the phosphorylation of FgSR. The authors concluded that FgSsk2 co-localized with FgSR to the nucleus based on Fig. 3d. However, Figure 3d clearly showed that FgSsk2 localized to the cytoplasm and FgSR localized to the nucleus under normal growth conditions. (Without activation by stresses, FgSsk2 is likely not in the nucleus) The authors used the same culture conditions for co-IP assays. It is questionable how can the authors detect such a strong interaction between FgSsk2 and FgSR by co-IP assays even though they differed so significantly in subcellular localization. Will stress or fungicide treatment affect their interactions? The authors should use BiFC assays to show their co-localization. (BiFC was used to show the dimerization of FgSR) Line 226-228. This conclusion about phosphorylation of FgSR by FgSsk2 is inappropriate. The authors did not present any direct evidence to show that FgSsk2 phosphorylates FgSR. The effect of FgSsk2 deletion on FgSR phosphorylation could be indirect because the normal function of FgSsk2 is in the Ssk2-Pbs2-Hog1 pathway. Is the activated or unactivated FgSsk2 that phosphorylates FgSR? Does the role of FgSsk2 in the activation of FgSR involve Pbs2 and Hog1? The authors could test this hypothesis by doing similar assays with the downstream MAPKK or MAPK mutants.

3. Regarding the putative cis element identified and characterized in this study, to this reviewer, the authors did not present convincing or direct evidence to show it is the binding site of FgSR. The in vivo effects of their deletion or mutation in the Cyp51A promoter may be caused by other

factors that bind to or recognize these elements (not by FgSR per se). The authors should conduct EMSA assays to show the direct binding of FgSR to these cis elements and mutational effects. Also, why only five of these genes, not majority of the sterol synthesis related genes, have these elements? (I don't know how reliable the CHIP-seq data generated by the authors)

4. I am also concerned with the five phosphorylation sites described in the manuscript. It was not clearly described how the authors identified these five sites. Is there any way to predict phosphorylation sites of MAPKKK on non-MAPKK targets? Also, because of the differences between FgSsk2 deletion and 5A mutations, the authors should do mutations at individual sites to identify the FgSsk2 phosphorylation sites. (Deletion of FgSSK2 had no effect on tebuconazole induced expression of Cyp51A but five phosphorylation sites were important for that). Another experiment is to generate the 5D mutations to mimic activation.

Regarding Figure 3h, if the authors could identify the FgSR phosphorylation sites by phosphoproteomics analysis, they should be able to use the same method to identify changes in the phosphorylation sites of FgSR in the FgSsk2 deletion mutant.

5. *Fusarium graminearum* has two SREBP genes and four Upc2 orthologs. It is possible that some of these genes may have overlapping functions. Based on phenotypical characterizations of the single mutants of these genes, the authors could not conclude that these genes are not important for regulating sterol synthesis. In fact, the FgSR deletion mutant was only reduced approximately 50% in growth rate under normal growth conditions, suggesting FgSR is not an important regulator of sterol synthesis because of the essential functions of ergosterol.

This transcription factor FgSR has been characterized in an earlier study (Son et al., 2011. PLoS Pathogens). Deletion of this transcription factor had pleiotropic defects, including increased sensitivity to oxidative stress and reduced virulence.

The authors identified approximately 50 putative FgSR-interacting proteins in yeast two-hybrid assays. FgSsk2 and Swp71 were just two of them. Somehow, both of them are related to sterol synthesis. ???

Minor points:

Fig. 3B. It appears to this reviewer that the images for the negative control and FgSR or FgSR5A in Fig. 3b were not the original images or images of low quality.

Based on Fig. 8 Sordariomycetes and Leotiomyces differ from other lower fungi and basidiomycetes or animals in the key regulator of sterol biosynthesis. Evolution?

Deletion of ML likely affects protein folding, which may affect many things other than dimerization. Which domain is responsible for DNA binding or binding to the cis elements identified in this study?

Reviewer #3 (Remarks to the Author):

NCOMMS-18-00653

This manuscript deals with the major plant pathogen *Fusarium graminearum* and the discovery of a novel regulator of sterol biosynthesis. From mammals to yeast, sterol biosynthesis is regulated by a rather conserved transcription factor called Upc2/SREBP. In this manuscript, a so far not described transcription factor FgSR is described that provides a novel mechanism for regulating sterol biosynthesis. Most importantly, this regulation differs very much from the regulation by the conserved SREBP/Upc2 orthologs. For example, FgSR is located in the nucleus independent of ergosterol starvation. Further FgSR protein levels do not alter by sterol treatment,

and the protein level bound to the promoters of target genes is not increased under sterol-deprived conditions. Finally, FgSR phosphorylation regulates its transcriptional activity via recruiting the SWI/SNF complex, which is responsible for chromatin remodeling. The manuscript is highly complex and provides a huge wealth of information. For example, the data are presented in nine figures that overall display 47 subfigures. However, the overall finding of this manuscript is restricted to a few ascomycetous groups, namely the Sordariomycetes and the Leotiomycetes. With this, a rather specific regulatory mechanism is described. Overall, it is a very ambitious paper and I have some points, which should be clarified before publication.

1. Line 189ff.: The authors claim that phosphorylated FgSR recruits the SWI/SNF complex. A major finding in this chapter is the interaction of FgSR with FgSSK2, encoding a major MMAPKK kinase. The authors predict that FgSR contains five predicted amino acid sites (line 217), however I do not see the real data for this. For example, Fig. 3h is a general phosphorylation experiment, and Fig. S6 only shows two phosphorylation sites as evidence by LC-MS/MS. Further in this chapter, the authors found an interaction of FgSwp73, a component of the SWI/SNF complex. This is solely based on a Y2H approach. They further provide other data, for example enrichment of FgArp9 at the FGCYP51A promoter or the enrichment of H3 at the promoter of FgCYP51A. All these are to my mind rather indirect evidences for the statements made by the authors that the SWI/SNF complex is recruited by FgSR and is involved in the transcriptional regulation of sterol biosynthesis genes (line 247-251).
2. Legend of Fig. 2 (line 832): PH-1 transformed with GFP served as a negative control. Maybe I do not understand this comment, but I do not see it in the figures.
3. Fig. 6A: The authors show the ChIPseq data for several of the ergosterol biosynthesis genes. However, three are missing. Explain why no data were received.
4. Fig. 7: The MEME analysis is shown for five genes. I do not understand why only five target genes were selected, although much more target genes were identified in the corresponding chapter (line 302-309).
5. Fig. 7: There are four promoter derivatives shown (P1-P4). However, P4 was not checked *in vivo* (Fig. 7C, D, E), this should be briefly explained. Further, some of the capital letters (A, B) are not explained in the legend.
6. The discussion contains many repetitions of the results, for example in line 494 or 482.
7. Further, in line 409 the authors mention that the binding-cis element for FgSR also exists in the promoters of other ascomycetes fungi. Has this been shown somewhere? Can they provide some references?

Minor points:

1. Line 859: "Predicted" instead of "predicated".
2. Fig. 2: It should read "promoter", not "prompter".
3. The designation of CYP51 is rather confusing. In line 151, it is called FgCYP51, in line 154 CYP51, and finally in line 171 FgCYP51s. Is this the plural?
4. Some letters in diverse figures are not abbreviated in the legends, e. g. Fig. 1-5.
5. In Fig. 5a some of the species were misspelled, e. g. *Podospora anserina* and *Aspergillus fumigatus*.

Author responses:

We thank editor and all reviewers for their greatly comments and suggestions, which have helped us to substantially improve the manuscript. Here are point-by-point to the comments raised by editor and the reviewers

Reviewer #1 (Remarks to the Author):

The manuscript by Liu et al describes the identification and detailed characterisation of a here-to unrecognised pathway that regulates sterol biosynthesis. After providing evidence that *Fusarium* homologues of known sterol regulatory pathways from other organisms did not have this role in *F. graminearum*, the authors undertook an elegant loss-of-function genetic screen to find transcription factors that contribute to basal tolerance to sterol biosynthesis inhibitors. The authors went on to characterise how the protein functions and the signalling and chromatin remodelling complexes involved in its regulation of sterol biosynthesis.

We thank the reviewer for the positive comments.

1. Although the bulk of the paper (and title) focuses on the role of the identified transcription factor and signalling pathways in sterol biosynthesis regulation the work provides tantalising evidence that the transcription factor has a broader role in regulation of transcription associated with the pathogenesis process. For this reason the title could be made slightly broader.

Re: Thank you for your suggestion. It is a good point. We added “pathogenicity” in the title.

2. Line 136: “restore the sensitivity of yeast Upc2 and Ecm22 mutants to azole compounds” could be interpreted that the Upc2 and Ecm22 mutants are actually resistant to these compounds which I suspect is not the intended meaning. Perhaps “restore” should be replaced with “complement”. Likewise in the supplementary figure 2 legend “restore sensitivity” implies the mutants are actually resistant.

Re: We replaced “restore” with “complement” in lines 141, 153, 344 and 1123.

3. Line 200: The link between the Y2H result and the Ssk2 mutant result does not read very well. Perhaps “Among them, we observed that the mutant of Fgssk2 also showed increased sensitivity to...” could be changed to “One interacting protein was the FgSsk2 MAPKK kinase and mutants in the encoding gene showed increase sensitivity to...”

Re: Thank you for your suggestion. We revised this sentence to make this part go smoothly (lines 205-207).

4. Line 206: In Figure 3e it appears the difference between expression of Cyp51A under tebuconazole treatment is only slightly reduced in the ssk2 mutant compared to Ph1. Should this sentence on line 206 read “to 89.2%”

Re: Thank you for your question. We carefully checked the original data and the result in Fig. 3f and line 214 is right. In Fig. 3f, we directly marked the relative expression of *FgCYP51A* in PH-1 under tebuconazole treatment with digit 78.05 to avoid the graph looking too high. The relative expression of *FgCYP51A* in FgSsk2 mutant is 8.47 in comparison with that of PH-1 without treatment. Therefore, deletion of *FgSSK2* decreased the expression level of *FgCYP51A* by 89.15% under tebuconazole treatment. To make the results clearly, we remade Fig. 3f.

5. It is less clear why the sensitivity to BOA was tested and how FgSR regulates the expression of the detoxification genes given these were not identified in the ChIP-seq data. Whilst the data are interesting around BOA a better justification of why these genes were investigated and an explanation of how FgSR might regulate their expression should be included.

Re: Our study found that deletion of *FgSR* reduced approximately 50% growth, but caused a dramatic decrease in virulence (Fig. 9a,b). We were therefore interested in exploring mechanism for the reduced virulence in Δ FgSR. Kettle and his colleagues have reported that degradation of the benzoxazolinone class of phytoalexins is important for virulence of *Fusarium*. Hence, we tested the sensitivity of FgSR to multiple phytoalexins including gramine, 2-benzoxazalinone and tryptamine, and found that Δ FgSR was specifically hypersensitive to BOA, but not to other

phytoalexins (Image A below).

The transcriptions of three genes (FgFDB1,-2, and -3) that were reported to be involved in BOA detoxification were down-regulated in Δ FgSR, but these three genes were not identified in the ChIP-Seq assay, indicating that FgSR might indirectly regulate expression of the three genes. Until now, the knowledge about the mechanism of BOA action and detoxification is very limited. To explore the potential target of BOA, we deleted 22 of the FgSR target genes identified by ChIP-Seq assay that are potential related to stress responses, and determined sensitivity of these mutants to BOA. But none of the mutants displayed changed sensitivity to BOA compared with the wild type (Image B below). Thus, molecular mechanisms of FgSR in regulating transcription of FgFDB1,-2,-3 remain unclear.

6. Figure 2: promoter is spelled incorrectly in the vertical axes of part b. It might also be useful to align the order of the different regions assayed in the promoters with the bars as they are currently in opposite orientation in the promoter diagram to the graphs below.

Re: Thank you for your reminder. We redrew the promoter diagram according the orders of ChIP-qPCR assays (Fig. 2b). And we corrected “promoter” in Fig. 2b.

7. In figure 2a, delta-erg4 induced ergosterol deficiency clearly induces cyp51a expression. However, in Supp Fig 5, cyp51a does not seem to be induced in the delta-erg4::FgSR-GFP strain which should functionally equivalent to the delta-erg4 strain used in figure 2a. Can the reason for this discrepancy be explained?

Re: Thank you very much for the point. Erg4 is the last enzyme catalyzing ergosterol biosynthesis and we used Δ FgErg4 to mimic the ergosterol-absent condition. You are right. We made a mistake in Fig. S5a. The transcription of *FgCYP51A* was induced by FgErg4 deletion as well as tebuconazole treatment. *FgCYP51A* was equally induced in Δ FgErg4 and Δ FgErg4::FgSR-GFP. Please see the revised Fig. S5a.

8. Line 408: vegetative not vegetable

Re: Thank you for your patience and kind reminder. We corrected this word (line 511).

9. The statistical analyses have been performed with quantitative data but the tests used to identify groups that differ (defined on bar charts by letters) have not been explained in the figure legends or the methods. Having said that the data is still very convincing so this is just a technicality.

Re: We added the information for statistical analysis in corresponding figure legends.

Reviewer #2 (Remarks to the Author):

Fusarium graminearum is an important plant pathogenic fungus. In this study, the authors showed that six orthologs of known regulators of ergosterol biosynthesis were dispensable. The FgSR mutant was then found to have increased sensitivity to tebuconazole. Enrichment of FgSR at the promoter of Cyp51A/B was observed. The authors then screened yeast two-hybrid library and identified FgSsk2 and Swp73 as two of the FgSR-interacting proteins. Five putative phosphorylation sites were identified and characterized in FgSR. Mutations at these five sites affected the interaction of FgSR with Swp73. The authors then showed that the SNF complex may

be recruited to the Cyp51A promoter via FgSR. They also found that FgSR was enriched in other genes related to ergosterol synthesis by ChIP-seq. Putative FgSR binding sites were identified. Interestingly, FgSR proteins were only found in Sordariomycetes and Leotiomycetes. Based on these data, the authors proposed that FgSR recruits the chromatin remodeling factors to regulate fungal sterol biosynthesis. To this reviewer, some of the conclusions were not well-supported by data presented in this manuscript.

Major concerns:

1. The subcellular localization and enrichment of FgSR on the CYP51 promoters were not affected by tebuconazole treatment. One major experimental evidence to show that phosphorylation of FgSR recruited the SNF complex to these promoters is that 5A mutation affected the interaction between FgSR and Swp73. Because this result is critical to the conclusions of this manuscript, it is important to prove the interaction between FgSR and Swp73 and the effect of 5A mutations on their interactions by co-IP assays and BiFC assays. A simple yeast two-hybrid assay is not sufficient. Although FgSsk2 was dispensable for tebuconazole induced enrichment, Swp73 and Arp9 were. Is that somewhat contradictory? According to the descriptions, phosphorylation of FgSR by FgSsk2 and other kinase is important for recruiting the SNF complex to the FgSR bound promoters.

Re: We conducted the Co-IP assays to detect the interaction of FgSR and FgSwp73. As shown in Fig. 4b, FgSR interacted with FgSwp73, and this interaction was enhanced under tebuconazole treatment (lines 254, 255). As you suggested, we performed BiFC assay to verify the interaction of FgSwp73 and FgSR, but did not observe the interaction signal. Previous studies have shown that BiFC assays may give false negatives, which may result from the conformation or activity of the tagged proteins are changed by tags (Kerppola, 2008; Lee et al, 2011). In this study, FgSsk2 phosphorylates FgSR. We therefore further tested the interaction strength of FgSsk2 and FgSR by Co-IP assays. As shown in Fig. 3c, this interaction was enhanced by tebuconazole treatment (lines 211, 212). Consistently, the high levels of the phosphorylated isoforms of FgSR in the wild type by tebuconazole treatment were

reduced in *FgSSK2* mutant (Fig. 3f). In addition, we found that phosphorylation of FgSR was critical for its interaction with FgSwp73, and the enrichment of FgSwp73 and FgArp9 at the promoter of *FgCYP51A* (Fig. 4).

Lee, O. H., Kim, H., He, Q., Baek, H. J., Yang, D., Chen, L. Y., ... & Songyang, Z. (2011). Genome-wide YFP fluorescence complementation screen identifies new regulators for telomere signaling in human cells. *Molecular & Cellular Proteomics*, 10(2), M110.001628.

Kerppola, T. K. (2008). Bimolecular fluorescence complementation (BiFC) analysis as a probe of protein interactions in living cells. *Annual Review of Biophysics and Biomolecular Structure*, 37, 465-487

2. I have serious doubt about the interaction of FgSsk2 with FgSR and the direct role of FgSsk2 in the phosphorylation of FgSR. The authors concluded that FgSsk2 co-localized with FgSR to the nucleus based on Fig. 3d. However, Figure 3d clearly showed that FgSsk2 localized to the cytoplasm and FgSR localized to the nucleus under normal growth conditions. (Without activation by stresses, FgSsk2 is likely not in the nucleus) The authors used the same culture conditions for co-IP assays. It is questionable how can the authors detected such a strong interaction between FgSsk2 and FgSR by co-IP assays even though they differed so significantly in subcellular localization. Will stress or fungicide treatment affect their interactions? The authors should use BiFC assays to show their co-localization. (BiFC was used to show the dimerization of FgSR)

Line 226-228. This conclusion about phosphorylation of FgSR by FgSsk2 is inappropriate. The authors did not present any direct evidence to show that FgSsk2 phosphorylates FgSR. The effect of FgSsk2 deletion on FgSR phosphorylation could be indirect because the normal function of FgSsk2 is in the Ssk2-Pbs2-Hog1 pathway. Is the activated or unactivated FgSsk2 that phosphorylates FgSR? Does the role of FgSsk2 in the activation of FgSR involves Pbs2 and Hog1?. The authors could tested this hypothesis by doing similar assays with the downstream MAPKK or MAPK mutants.

Re: Thank you for insightful suggestions. The IP samples of Co-IP assays were enriched by GFP-Trap agarose beads. Thus, the interaction signal could be enlarged in Co-IP assay. Regarding the subcellular localization of FgSsk2, because the RFP

fluorescence signal of PH-1::FgSsk2-RFP was weak, and the native untagged FgSsk2 may affect localization of the FgSsk2-RFP that was introduced into the wild type (in the original Fig. 3d), we knocked out the native *FgSSK2* in this revision, and found that the FgSsk2-RFP obviously localized to both the nucleus and cytosol in Δ FgSsk2::FgSsk2-RFP strain (Fig. 3d). Again, we conducted BiFC, and did not observe the interaction signal. Therefore, we further performed immunofluorescence assays, and found that both FgSsk2 and FgSR are clearly present in the nucleus (Fig. 3e). Moreover, the Co-IP assay showed that FgSR-FgSsk2 interaction was enhanced by tebuconazole treatment (Fig. 3c). These results indicate that FgSsk2 is able to interact with FgSR in the nucleus.

As you suggested, we tested the interaction of FgSR with FgPbs2 and FgHog1. The yeast two-hybrid assays showed that FgSR does not interact with FgPbs2 and FgHog1, but with FgSsk2 (Fig. S6). These results indicate that FgSsk2, but not FgPbs2 and FgHog1, is involved in phosphorylating FgSR (Fig. S6; lines 219-223).

3. Regarding the putative cis element identified and characterized in this study, to this reviewer, the authors did not present convincing or direct evidence to show it is the binding site of FgSR. The in vivo effects of their deletion or mutation in the *Cyp51A* promoter may be caused by other factors that bind to or recognize these elements (not by FgSR per se). The authors should conduct EMSA assays to show the direct binding of FgSR to these cis elements and mutational effects. Also, why only five of these genes, not majority of the sterol synthesis related genes, have these elements? (I don't know how reliable the ChIP-seq data generated by the authors)

Re: We have conducted the yeast one-hybrid and the result showed that FgSR is able to bind the promoter of *FgCYP51A* via two repeats of CGAA element directly. Many previously reports have shown that the yeast one-hybrid is a reliable approach to determine direct binding of a protein to DNA element in a heterologous system (Hens *et al*, 2012; Wanke *et al*, 2009).

In addition, we also tried to perform EMSA, but failed in purification of FgSR-His via the *E. coli* expression system because FgSR-His was mainly present the

inclusion body. We tried to optimize different conditions, such as temperature, prokaryotic expression vectors, but FgSR-His was still present the inclusion body (Image below).

In this study, the ChIP-Seq analysis provided a putative *cis*-element (revised Fig. S8e). To get an accurate *cis*-element that is bound by FgSR. We selected five ergosterol-related genes, which were down-regulated in RNA-Seq and targeted by FgSR in the ChIP-Seq assay. By using the MEME (Motif-based sequence analysis tools), we found that the promoters of all these five genes contain a predicted 16-bp *cis*-element bearing two CGAA-repeated sequences (Fig. 7a). Further analysis revealed that this *cis*-element existed in the promoters of 64 (54%) FgSR target genes identified from the ChIP-Seq assay including 20 sterol biosynthesis genes (Fig. 6a; Table S3).

Hens, K., Feuz, J. D., & Deplancke, B. (2012). A high-throughput gateway-compatible yeast one-hybrid screen to detect protein-DNA interactions. *Methods in Molecular Biology*, 786(786), 335-355.

Wanke, D., & Harter, K. (2009). Analysis of plant regulatory DNA sequences by the yeast-one-hybrid assay. *Methods in Molecular Biology*, 479, 291-309.

4. I am also concerned with the five phosphorylation sites described in the manuscript. It was not clearly described how the authors identified these five sites. Is there any way to predict phosphorylation sites of MAPKKK on non-MAPKK targets? Also, because of the differences between FgSsk2 deletion and 5A mutations, the authors should do mutations at individual sites to identify the FgSsk2 phosphorylation sites.

(Deletion of FgSSK2 had no effect on tebuconazole induced expression of Cyp51A but five phosphorylation sites were important for that). Another experiment is to generate the 5D mutations to mimic activation.

Regarding Figure 3h, if the authors could identified the FgSR phosphorylation sites by phosphoproteomics analysis, they should be able to use the same method to identify changes in the phosphorylation sites of FgSR in the FgSsk2 deletion mutant.

Re: Five sites of FgSR were predicated by using the program NetPhos (<http://www.cbs.dtu.dk/services/NetPhos/>). Further, liquid chromatography tandem mass spectrometry (LC-MS/MS) analysis verified that two of them were phosphorylated under tebuconazole treatment. Hence, we further studied the function of these five sites.

Based on your suggestion, we constructed the 5D mutant (phosphorylated mutant) and found that FgSR^{5D} could restore the sensitivity of Δ FgSR to tebuconazole and the transcription of *FgCYP51A* with tebuconazole treatment (Fig.3a,f). We also obtained the mutants containing individual phosphorylation site mutated from S/T to A (dephosphorylated mutation), and found these mutants did not exhibit increased sensitivity to tebuconazole, indicating the function of these five phosphorylation sites may be redundant. Please see lines 241-245, and Fig. S7b.

In Fig. 3f, we directly marked the relative expression of *FgCYP51A* in PH-1 under tebuconazole treatment with digit 78.05 to avoid the graph looking too high. The relative expression of *FgCYP51A* in FgSsk2 mutant is 8.47 in comparison with that of PH-1 without treatment. Therefore, deletion of *FgSSK2* decreased the expression level of *FgCYP51A* by 89.15% under tebuconazole treatment. We reconstructed Fig. 3f to make the results clearly. Furthermore, to identify the sites of FgSR potentially phosphorylated by FgSsk2, we detected the phosphoproteomics in Δ FgSsk2::FgSR-GFP treated by tebuconazole in two independent LC-MS/MS analyses. The result showed that S305 of FgSR site was not phosphorylated in the mutant, but it was phosphorylated in the wild type.

5. *Fusarium graminearum* has two SREBP genes and four Upc2 orthologs. It is possible that some of these genes may have overlapping functions. Based on phenotypical characterizations of the single mutants of these genes, the authors could not conclude that these genes are not important for regulating sterol synthesis. In fact, the FgSR deletion mutant was only reduced approximately 50% in growth rate under normal growth conditions, suggesting FgSR is not an important regulator of sterol synthesis because of the essential functions of ergosterol.

This transcription factor FgSR has been characterized in an earlier study (Son et al., 2011. PLoS Pathogens). Deletion of this transcription factor had pleiotropic defects, including increased sensitivity to oxidative stress and reduced virulence.

The authors identified approximately 50 putative FgSR-interacting proteins in yeast two-hybrid assays. FgSsk2 and Swp71 were just two of them. Somehow, both of them are related to sterol synthesis?

Re: Thanks for your constructive comments. We conducted additional experiments to confirm that SREBP and Upc2 orthologs were not involved in ergosterol biosynthesis. First, we constructed the double deletion mutants (Δ FgSre1/FgSre2, Δ FgUpc2A/B, Δ FgUpc2A/C, Δ FgUpc2A/D, Δ FgUpc2B/C, Δ FgUpc2B/D, Δ FgUpc2C/D), and found that they did not alter sensitivity to tebuconazole compared with wild type (Fig. S1b). Second, *FgCYP51A* still could be up-regulated by tebuconazole treatment in the SREBP and UPC2 single or double deletion mutants (Fig. S1c). Third, the transcription of each SREBP or Upc2 ortholog was not altered by tebuconazole treatment or deletion of another SREBP or Upc2 ortholog (Fig. S1d). Taken together, SREBP and Upc2 orthologs in *F. graminearum* are not involved in ergosterol biosynthesis.

Erg4 is the last enzyme catalyzing ergosterol biosynthesis. Our previous study has reported that *FgERG4* deletion mutant displayed 50% growth relative to the wild type although the mutant could not synthesize ergosterol (Liu et al, 2013). In *S. cerevisiae*, deletion of *ERG4* also is viable and Δ ERG4 shows similar growth compared with the wild type although Δ ERG4 cannot biosynthesize ergosterol (Zweytick et al, 2001). Moreover, deletion of the master sterol regulator in

Aspergillus fumigatus or *Cryptococcus neoformans* only caused a slight decrease in growth under the normal conditions (Willger et al, 2008; Chun et al, 2009). Taken together, it is reasonable to draw the conclusion that FgSR is the master regulator of sterol biosynthesis in *F. graminearum* although Δ FgSR only exhibited approximately 50% growth relative to the wild type under the normal conditions.

We apologized for making a mistake in numbering FgSR-interacting proteins obtained by screening cDNA library of *F. graminearum*. In fact, total 67 FgSR-interacting proteins were identified. Among them, 44 potentially ergosterol-related proteins were knocked out. Determining the sensitivity of 44 mutants to tebuconazole, we found that 41 mutants showed similar sensitivity, only kinase FgSsk2 (FGSG_00408), kinase FgCdc15 (FGSG_10381) and FgSR (FGSG_01176) deletion mutants displayed increased sensitivity in comparison with the wild type (Image below). Further, yeast two-hybrid assays confirmed that FgSR interacted with the full length of FgSsk2, but not the full length of FgCdc15. Therefore, we focused on the function of FgSsk2 for regulating FgSR in this study.

- Davies, B. S., Wang, H. S., & Rine, J. (2005). Dual activators of the sterol biosynthetic pathway of *Saccharomyces cerevisiae*: similar activation/regulatory domains but different response mechanisms. *Molecular and cellular biology*, 25(16), 7375-7385.
- Zweytick, D., Hrastnik, C., Kohlwein, S. D., & Daum, G. (2000). Biochemical characterization and subcellular localization of the sterol c-24(28) reductase, erg4p, from the yeast *Saccharomyces cerevisiae*. *Febs Letters*, 470(1), 83-7.
- Liu, X., Jiang, J., Yin, Y., & Ma, Z. (2013). Involvement of fgerg4 in ergosterol biosynthesis, vegetative differentiation and virulence in *Fusarium graminearum*. *Molecular Plant Pathology*, 14(1), 71.
- Willger, S. D., Puttikamonkul, S., Kim, K. H., Burritt, J. B., Grahl, N., Metzler, L. J., ... & Cramer Jr, R. A. (2008). A sterol-regulatory element binding protein is required for cell polarity, hypoxia adaptation, azole drug resistance, and virulence in *Aspergillus fumigatus*. *PLoS pathogens*, 4(11), e1000200.
- Chun, C. D., Liu, O. W., & Madhani, H. D. (2007). A link between virulence and homeostatic responses to hypoxia during infection by the human fungal pathogen *Cryptococcus neoformans*. *PLoS pathogens*, 3(2), e22.

Minor points:

1. Fig. 3B. It appears to this reviewer that the images for the negative control and FgSR or FgSR5A in Fig. 3b were not the original images or images of low quality.

Re: We replaced the images in Fig. 3b with images having high resolution from another repeated experiment.

2. Based on Fig. 8 Sordariomycetes and Leotiomycetes differ from other lower fungi and basidiomycetes or animals in the key regulator of sterol biosynthesis. Evolution? \

Re: Yes. It has been reported that *Fusarium graminearum* is located in the middle stage between *Saccharomycotina* and the higher fungi during evolution (Fitzpatrick et al, 2006; James et al, 2006). So it is reasonable to deduce that the FgSR may serve as an evolutionary intermediate sterol regulator from Upc2 and SREBP.

Fitzpatrick, D. A., Logue, M. E., Stajich, J. E., & Butler, G. (2006). A fungal phylogeny based on 42 complete genomes derived from supertree and combined gene analysis. *BMC evolutionary biology*, 6(1), 99.

James, T. Y., Kauff, F., Schoch, C. L., Matheny, P. B., Hofstetter, V., Cox, C. J., ... & Lumbsch, H. T. (2006). Reconstructing the early evolution of Fungi using a six-gene phylogeny. *Nature*, 443(7113), 818.

3. Deletion of ML likely affects protein folding, which may affect many things other than dimerization. Which domain is responsible for DNA binding or binding to the cis elements identified in this study?

Re: Thank you for your good suggestion. According to the NCBI (<https://www.ncbi.nlm.nih.gov/>) and SMART (<http://smart.embl-heidelberg.de/>) analysis, FgSR contains a zinc finger (ZF) domain which is a typical DNA binding domain (Lee et al, 1989; Persikov et al, 2015). We conducted experiments to study

the function of this zinc finger domain. As shown in revised Fig. 5d-f, the strain lacking the ZF domain (FgSR-C^{ΔZF}) displayed the defects in response to azole compounds, ergosterol biosynthesis, induction of *FgCYP51A* expression by tebuconazole treatment, and the binding ability to *FgCYP51A* promoter (lines 291-296). These results indicate that the zinc finger domain is responsible for DNA binding.

Previous studies have found that the dimerization domain of zinc finger transcription factor is primarily located in the C-terminal next to the DNA binding domain (Näär et al, 2009). In this study, deletion of the middle linker (ML) domain disrupted the formation of FgSR homodimer (Fig. 5a-c). Further, similar to ΔFgSR and ΔFgSR-C^{ΔZF}, the strain lacking the ML domain (ΔFgSR-C^{ΔML}) could not bind to the promoter of *FgCYP51A*, further displayed the defects in response to azole compounds, ergosterol biosynthesis, and induction of *FgCYP51A* expression by tebuconazole treatment (Fig. 5d-f). Moreover, ChIP-qPCR assay showed that similar to ΔFgSR-C^{ΔZF}, FgSR^{ΔML} could not bind to the promoter of *FgCYP51A* (Fig. 5h). Therefore, we proposed that the homodimerization of FgSR mediated by the ML domain is required for its binding to the target gene promoter (lines 290-300).

- Lee, M. S., Gippert, G. P., Soman, K. V., Case, D. A., & Wright, P. E. (1989). Three-dimensional solution structure of a single zinc finger DNA-binding domain. *Science*, 245(4918), 635-637.
- Persikov, A. V., Wetzel, J. L., Rowland, E. F., Oakes, B. L., Xu, D. J., Singh, M., & Noyes, M. B. (2015). A systematic survey of the Cys2His2 zinc finger DNA-binding landscape. *Nucleic acids research*, 43(3), 1965-1984.
- Näär, A. M., & Thakur, J. K. (2009). Nuclear receptor-like transcription factors in fungi. *Genes & Development*, 23(4), 419-32.

Reviewer #3 (Remarks to the Author):

This manuscript deals with the mayor plant pathogen *Fusarium graminearum* and the discovery of a novel regulator of sterol biosynthesis. From mammals to yeast, sterol biosynthesis is regulated by a rather conserved transcription factor called Upc2/SREBP. In this manuscript, a so far not described description factor FgSR is

described that provides a novel mechanism for regulating sterol biosynthesis. Most importantly, this regulation differs very much from the regulation by the conserved SREBP/Upc2 orthologs. For example, FgSR is located in the nucleus independent of ergosterol starvation. Further FgSR protein levels do not alter by sterol treatment, and the protein level bound to the promoters of target genes is not increased under sterol-deprived conditions. Finally, FgSR phosphorylation regulates its transcriptional activity via recruiting the SWI/SNF complex, which is responsible for chromatin remodeling. The manuscript is highly complex and provides a huge wealth of information. For example, the data are presented in nine figures that overall display 47 subfigures. However, the overall finding of this manuscript is restricted to a few ascomycetous groups, namely the Sordariomycetes and the Leotiomycetes. With this, a rather specific regulatory mechanism is described. Overall, it is a very ambitious paper and I have some points, which should be clarified before publication.

Thank you very much for the positive comments.

1. Line 189ff.: The authors claim that phosphorylated FgSR recruits the SWI/SNF complex. A mayor finding in this chapter is the interaction of FgSR with FgSSK2, encoding a mayor MMAPKK kinase. The authors predict that FgSR contains five predicted amino acid sites (line 217), however I do not see the real data for this. For example, Fig. 3h is a general phosphorylation experiment, and Fig. S6 only shows two phosphorylation sites as evidence by LC-MS/MS.

Re: Five phosphorylation sites of FgSR were predicated by using the program NetPhos (<http://www.cbs.dtu.dk/services/NetPhos/>). Furthermore, liquid chromatography tandem mass spectrometry (LC-MS/MS) analysis detected that two of them were phosphorylated under tebuconazole treatment. Phosphorylation was invertible and dynamic post-translational modification. Thus, we therefore mutated the five amino acid sites and detect the influence on the function of FgSR.

Based on your suggestion, we constructed the 5D mutant FgSR-C^{5D} (a mimic phosphorylated isoform) and found that the mutant could restore the sensitivity of Δ FgSR to tebuconazole and the transcription of *FgCYP51A* with tebuconazole

treatment (Fig.3a,f). We also obtained the mutants containing individual non-phosphorylated mutation (from S/T to A), and found these mutants did not exhibit increased sensitivity to tebuconazole, indicating the function of these five sites may be redundant. Please see lines 241-245, and Fig. S7b.

2. Further in this chapter, the authors found an interaction of FgSwp73, a component of the SWI/SNF complex. This is solely based on a Y2H approach. They further provide other data, for example enrichment of FgArp9 at the FGCYP51A promoter or the enrichment of H3 at the promoter of FgCYP51A. All these are to my mind rather indirect evidences for the statements made by the authors that the SWI/SNF complex is recruited by FgSR and is involved in the transcriptional regulation of sterol biosynthesis genes (line 247-251).

Re: The Co-IP assay was conducted to confirm that FgSR interacts with FgSwp73. Moreover, the Co-IP assay showed that the interaction of these two proteins was enhanced under tebuconazole treatment. We have added the results in the revised manuscript. Please see the revised Fig. 4b and lines 254, 255.

In few eukaryotic organisms, it has been reported that the transcription factor could recruit histone modification or chromatin remodeling complexes to regulate gene transcription. The recruitment processes were verified by comparing the enrichment of components of the complex at the target genes in the wild type and the mutant of transcription factor (Nguyen et al, 2017; Cheng et al, 2018; Proft et al 2002; Guan et al, 2011). Therefore, we detected the enrichment of FgArp9 at the *FgCYP51A* promoter in Δ FgSR to confirm the recruit of the SWI/SNF complex by FgSR, and we further detected the enrichment of H3 at the *FgCYP51A* promoter in Δ FgSR to confirm chromatin remodeling function of the SWI/SNF complex.

Nguyen, T. T., Savory, J. G., Brooke-Bisschop, T., Ringuette, R., Foley, T., Hess, B. L., ... & Lohnes, D. (2017). Cdx2 regulates gene expression through recruitment of Brg1-associated Switch-Sucrose Non-fermentable (SWI-SNF) chromatin remodeling activity. *Journal of Biological Chemistry*, 292(8), 3389-3399.

- Cheng, S., Tan, F., Lu, Y., Liu, X., Li, T., Yuan, W., ... & Zhou, D. X. (2018). WOX11 recruits a histone H3K27me3 demethylase to promote gene expression during shoot development in rice. *Nucleic Acids Research*.
- Proft, M., & Struhl, K. (2002). Hog1 kinase converts the Sko1-Cyc8-Tup1 repressor complex into an activator that recruits SAGA and SWI/SNF in response to osmotic stress. *Molecular cell*, 9(6), 1307-1317.
- Guan, B., Wang, T. L., & Shih, I. M. (2011). ARID1A, a factor that promotes formation of SWI/SNF-mediated chromatin remodeling, is a tumor suppressor in gynecologic cancers. *Cancer research*, 71(21), 6718-6727.

3. Legend of Fig. 2 (line 832): PH-1 transformed with GFP served as a negative control. Maybe I do not understand this comment, but I do not see it in the figures.

Re: In order to confirm that enrichment determination of FgSR-GFP the promoter of *FgCYP51A* is reliable, and not caused by the GFP protein, we determined the GFP enrichment at *FgCYP51A* promoter in the negative control strain PH-1::GFP, and found that the GFP enrichment was undetectable in the promoter of *FgCYP51A* (down and right panel in Fig. 2b), indicating that GFP cannot bind the *FgCYP51A* promoter.

4. Fig. 6A: The authors show the ChIP-seq data for several of the ergosterol biosynthesis genes. However, three are missing. Explain why no data were received.

Re: According to the ChIP-Seq analysis, FgSR showed the significant enrichment at the promoters of 20 ergosterol biosynthesis genes, but not at the promoters of 8 genes in ergosterol biosynthetic pathway. We therefore did not display ChIP-Seq data for the eight genes in Fig. 6a. Similarly, in *Saccharomycotina* and *Aspergillus fumigatus*, the sterol regulators are unable to bind all genes in sterol biosynthesis pathway (Maguire et al, 2014; Chung et al, 2014).

Maguire, S. L., Wang, C., Holland, L. M., Brunel, F., Neuvéglise, C., & Nicaud, J. M., *et al.* (2014). Zinc finger transcription factors displaced srebp proteins as the major sterol regulators during *Saccharomycotina* evolution. *PLoS Genetics*, 10(1), e1004076.

Chung, D., Barker, B. M., Carey, C. C., Merriman, B., Werner, E. R., & Lechner, B. E., *et al.* (2014). Chip-seq and in vivo transcriptome analyses of the *Aspergillus fumigatus* SREBP SrbA reveals a new regulator of the fungal hypoxia response and virulence. *Plos Pathogens*, 10(11), e1004487.

5. Fig. 7: The MEME analysis is shown for five genes. I do not understand why only

five target genes were selected, although much more target genes were identified in the corresponding chapter (line 302-309).

Re: Indeed, the ChIP-Seq analysis provided a putative *cis*-element (revised Fig. S8e). To get an accurate *cis*-element that was bound by FgSR. We selected five ergosterol-related genes, which were down-regulated in RNA-seq and the promoters of these genes are bound by FgSR in ChIP-Seq assay. Thus, by using the MEME (Motif-based sequence analysis tool) program, we found the promoters of all these five genes have the 16-bp *cis*-element containing two CGAA-repeated sequences (Fig. 7a), which is consistent with the putative *cis*-element identified from ChIP-Seq analysis. In addition, this element is present in the promoters of 64 (54%) FgSR target genes.

6. Fig. 7: There are four promoter derivatives shown (P1-P4). However, P4 was not checked *in vivo* (Fig. 7C, D, E), this should be briefly explained. Further, some of the capital letters (A, B) are not explained in the legend.

Re: Thank you for your suggestion. The P4 only contains two repeats of the *cis*-element, and did not contain the remaining sequence of *FgCYP51A* promoter that is bound by RNA polymerase II complex or other transcription initiation-related factors. Thus, the P4 cannot work alone in activating gene transcription. Thus, the functions of P4 in regulating gene expression was not checked *in vivo* (Fig, 7C, D, E)

We added the explanation for capital letters A and B in lines 1020 and 1030.

7. The discussion contains many repetitions of the results, for example in line 494 or 482.

Re: Thanks a lot. We revised the discussion part in lines 522-527.

8. Further, in line 409 the authors mention that the binding-*cis* element for FgSR also exists in the promoters of other ascomycetes fungi. Has this been shown somewhere? Can they provide some references?

Re: The binding *cis*-elements of FgSR homologs from other ascomycetes were identified by the MEME software using the promoters of *ERG11* homologs as shown in Fig. 8 and Table S5. No studies reported the binding *cis*-elements of FgSR homologs in eukaryotes until now.

Minor points:

1. Line 859: “Predicted” instead of “predicated”.

Re: We have made the correction in lines 880 and 913. Thank you.

2. Fig. 2: It should read “promoter”, not “prompter”.

Re: We have made the correction in Fig. 2. Thank you.

3. The designation of CYP51 is rather confusing. In line 151, it is called FgCYP51, in line 154 CYP51, and finally in line 171 FgCYP51s. Is this the plural?

Re: In the yeast, there are one *CYP51* gene (also named *ERG11*), but *F. graminearum* contains three paralogous of *CYP51* gene, designated as *FgCYP51A*, *-B*, *-C*. Hence, the “*FgCYP51s*” indicates the plural. We replaced “*FgCYP51s*” with “*FgCYP51* genes” in lines 176, 178, 179 and 192.

4. Some letters in diverse figures are not abbreviated in the legends, e. g. Fig. 1-5.

Re: Thank you for your reminder. We checked all the figure legends and used abbreviated form for “Figure 2” in lines 904, 933, 946, 978, 990, 1024 and 1174.

5. In Fig. 5a some of the species were misspelled, e. g. *Podospora anserina* and *Aspergillus fumigatus*.

Re: Thanks a lot for your patience and reminder. We corrected the species names in Fig. 8a.

Reviewers' comments:

Reviewer #1 (Remarks to the Author):

I am satisfied all my original comments have been addressed. I wish to congratulate the authors on a such an interesting and very thorough study.

Reviewer #2 (Remarks to the Author):

First, I commend the authors' efforts to address my concerns with additional experiments, revised figures, and detailed responses. Nevertheless, some of my concerns are not fully addressed.

1. I still have doubts about the interaction of FgSsk2 with FgSR and the direct role of FgSsk2 in the phosphorylation of FgSR. For the direct interaction between FgSsk2 and FgSR, in the previous submission, the authors showed the localization of FgSsk2 to the cytoplasm and localization of FgSR to the nucleus although co-IP data showed they interacted in hyphae cultured under normal conditions. In the revised manuscript, the authors generated a new transformant and showed strong FgSsk2-RFP signals in hyphae cultured under normal conditions. The explanation that the endogenous FgSsk2 may interfere the localization of FgSsk2-RFP in the previous submission did not make sense to me (It should not because of co-IP with the epitope tag). More importantly, my understanding is that localization of activated MAP kinases or MAP kinase complexes to the nucleus is a dynamic process. Assuming the localization of FgSsk2 to the nucleus with FgPbs2 and FgHog1, should the localization of FgSsk2-RFP to the nucleus occurred dynamically (in-and-out, transiently) ONLY after this pathway was activated? Why do all the nuclei have strong RFP signals under normal culture conditions, assuming FgSsk2 and its downstream kinases are not activated? Also, if this assumption of the FgSsk2-FgPbs2-FgHog1 complex is correct, should the authors be able to detect the interaction of FgPbs2 or FgHog1 with FgSR if they could detect the FgSsk2-FgSR interaction by co-IP?

It is puzzling to this reviewer that FgSsk2-RFP localized to the nucleus under normal culture conditions (not activated). Should FgSsk2, FgPbs1, or FgHog1 be mainly in the cytoplasm before this pathway is activated? Is it possible that the localization of FgSsk2-RFP to the nucleus under normal culture conditions is due to overexpression of the fusion construct? Along the same line, the strong interaction between FgSR and FgSsk2 detected by co-IP may be due to the over-expression of FgSR fusion construct. The interaction between a kinase and its substrate is also a transient process.

It also sounds somewhat odd for the explanation given by the authors about the BiFC assays. They showed the dimerization of FgSR by BiFC and localization of FgSsk2-RFP to the nucleus. Does that mean fusion with split GFP or RFP had no effect on their localizations to the nucleus? Somehow, the interaction of Swp71 with FgSR also could not be visualized by BiFC.

2. Regarding phosphorylation of FgSR by FgSsk2, the authors still did not present any direct evidence to show that FgSsk2 directly phosphorylates FgSR in the revised manuscript. The effect of FgSsk2 deletion on FgSR phosphorylation could be indirect because the normal function of FgSsk2 is in the Ssk2-Pbs2-Hog1 pathway. Deletion of FgSsk2 will affect the activation of FgPbs2 and FgHog1. Is the FgHog1 or FgPbs2 mutant defective in FgSR activation?

Based on a quick literature search, the Fgssk2, FgPbs2, and FgHog1 mutants had the same phenotypes. If these mutants of the same MAP kinase pathway phenocopied each other, how can FgSSK2 have a direct phosphorylation target as important as FgSR in *F. graminearum*? The authors may need to discuss about this point.

3. Regarding the putative cis element recognized by FgSR, I am not sure whether yeast one-hybrid data are good enough as direct evidence. Unfortunately, the authors had troubles to generate

recombinant FgSR proteins for EMSA assays.

Reviewer #3 (Remarks to the Author):

Nature Communications manuscript NCOMMS-18-00653A

The authors provide a revised version of a previous manuscript and addressed all concerns, made in the reviews for the initial manuscript.

I appreciate that major concerns regarding phosphorylation sites of the transcription factor were addressed. In particular, the predicted phosphorylation sites were identified using NetPhos. And this explains in part the follow-up experiments. (see also the critics of reviewer #2, point2)

Another concern was related to the identified cis-element that putatively was bound by transcription factor FgSR. Although the authors gave a lot of significant answers, concerning this cis-element, an important experiment is still missing. They were unable to demonstrate the binding of the cis-element by FgSR using an EMSA assay (see reviewer #2, point 3 and the corresponding answer).

All other points were addressed and most of the sloppy spelling mistakes were corrected.

Response to reviewers

We thank all reviewers for their greatly comments and suggestions, which have helped us to substantially improve the manuscript. Here is point-by-point to the comments raised by the reviewers.

Reviewer #1 (Remarks to the Author):

I am satisfied all my original comments have been addressed. I wish to congratulate the authors on such an interesting and very thorough study.

We thank the reviewer for the positive comments.

Reviewer #2 (Remarks to the Author):

First, I commend the authors' efforts to address my concerns with additional experiments, revised figures, and detailed responses. Nevertheless, some of my concerns are not fully addressed.

1. I still have doubts about the interaction of FgSsk2 with FgSR and the direct role of FgSsk2 in the phosphorylation of FgSR. For the direct interaction between FgSsk2 and FgSR, in the previous submission, the authors showed the localization of FgSsk2 to the cytoplasm and localization of FgSR to the nucleus although co-IP data showed they interacted in hyphae cultured under normal conditions. In the revised manuscript, the authors generated a new transformant and showed strong FgSsk2-RFP signals in hyphae cultured under normal conditions. The explanation that the endogenous FgSsk2 may interfere the localization of FgSsk2-RFP in the previous submission did not make sense to me (It should not because of co-IP with the epitope tag). More importantly, my understanding is that localization of activated MAP kinases or MAP kinase complexes to the nucleus is a dynamic process. Assuming the localization of FgSsk2 to the nucleus with FgPbs2 and FgHog1, should the localization of FgSsk2-RFP to the nucleus occurred dynamically (in-and-out, transiently) ONLY after this pathway was activated? Why do all the nuclei have strong RFP signals under normal culture conditions, assuming FgSsk2 and its downstream kinases are not activated? Also, if this assumption of the FgSsk2-FgPbs2-FgHog1 complex is correct, should the

authors be able to detect the interaction of FgPbs2 or FgHog1 with FgSR if they could detect the FgSsk2-FgSR interaction by co-IP?

It is puzzling to this reviewer that FgSsk2-RFP localized to the nucleus under normal culture conditions (not activated). Should FgSsk2, FgPbs1, or FgHog1 be mainly in the cytoplasm before this pathway is activated? Is it possible that the localization of FgSsk2-RFP to the nucleus under normal culture conditions is due to overexpression of the fusion construct? Along the same line, the strong interaction between FgSR and FgSsk2 detected by co-IP may be due to the over-expression of FgSR fusion construct. The interaction between a kinase and its substrate is also a transient process.

It also sounds somewhat odd for the explanation given by the authors about the BiFC assays. They showed the dimerization of FgSR by BiFC and localization of FgSsk2-RFP to the nucleus. Does that mean fusion with split GFP or RFP had no effect on their localizations to the nucleus? Somehow, the interaction of Swp71 with FgSR also could not be visualized by BiFC.

Regarding phosphorylation of FgSR by FgSsk2, the authors still did not present any direct evidence to show that FgSsk2 directly phosphorylates FgSR in the revised manuscript. The effect of FgSsk2 deletion on FgSR phosphorylation could be indirect because the normal function of FgSsk2 is in the Ssk2-Pbs2-Hog1 pathway. Deletion of FgSsk2 will affect the activation of FgPbs2 and FgHog1. Is the FgHog1 or FgPbs2 mutant defective in FgSR activation? Based on a quick literature search, the Fgssk2, FgPbs2, and FgHog1 mutants had the same phenotypes. If these mutants of the same MAP kinase pathway phenocopied each other, how can FgSSK2 have a direct phosphorylation target as important as FgSR in *F. graminearum*? The authors may need to discuss about this point.

Re: Thank you very much for your insightful suggestions. A previous study has well characterized the FgSsk2-FgPbs2-FgHog1 pathway, and found that Δ FgSsk2 showed minor phenotypic differences in colony morphology, surface hydrophobicity and regulating Mgv1 and Gpmk1 phosphorylation compared with Δ FgHog1 and Δ FgPbs2 mutants (Wang et al., PLoS Pathogens, 2011; Zheng et al., PLoS One, 2012). Hence,

we thought that FgSsk2 may perform biological function which is independent on FgPbs2 and FgHog1, and therefore did not pay an attention on phosphorylation of FgSR by FgHog1 in our previous submission. Moreover, as shown in the Wang's paper (PloS Pathogens, 2011) and Zheng's paper (PLoS One, 2012), FgSsk2, FgHog1 and FgPbs2 mutant showed defective growth under the normal condition, and the FgHog1 could be phosphorylated in the normal condition. In addition, germlings of the FgHog1-GFP transformant had detectable GFP signals in the nucleus without NaCl treatment (Fig. 7 in Zheng's paper), so we think it is reasonable to detect the interaction of FgSR-FgSsk2 and the nucleus localization of FgSsk2 in the normal condition in the previous revision of our manuscript.

As you suggested, in this resubmission, we tested the sensitivity of Δ FgPbs2 and Δ FgHog1 to tebuconazole and found that similar to Δ FgSsk2, both Δ FgPbs2 and Δ FgHog1 also exhibited increased sensitivity to tebuconazole (Fig. 3a). Consistent with this observation, deletion of FgSsk2, FgPbs2 or FgHog1 dramatically decreased the expression level of *FgCYP51A* under tebuconazole treatment, although the deletion did not affect the enrichment of FgSR at the promoter of *FgCYP51A* (Fig. 3b; Supplementary Fig. 6d). To explore the relationship between HOG pathway and tebuconazole sensitivity, we found that tebuconazole treatment increased the phosphorylation level and nuclear localization of FgHog1 (Fig. 3c,d). Moreover, phos-tag assays revealed that the amount of the dephosphorylated isoforms of FgSR in Δ FgSsk2, Δ FgPbs2 and Δ FgHog1 was significantly higher than that in the wild type under tebuconazole treatment (Fig. 3e), indicating that tebuconazole may activate FgSR via HOG pathway-mediated phosphorylation. Subsequently, we found that FgSR interacts with the MAP kinase FgHog1, and this interaction is dependent on FgHog1 phosphorylation mediated by FgSsk2 and enhanced by tebuconazole treatment using Co-IP and BiFC assays (Fig. 3f,g). Further, FgSR directly interacts with FgHog1 in the pull down assay, and is phosphorylated by FgHog1 directly *in vitro* (Fig. 3h,i). Taken together, these results indicate that tebuconazole treatment activates the HOG pathway, and the activated FgHog1 further phosphorylates FgSR in nucleus to regulate *FgCYP51A* expression.

2. Regarding the putative *cis* element recognized by FgSR, I am not sure whether yeast one-hybrid data are good enough as direct evidence. Unfortunately, the authors had troubles to generate recombinant FgSR proteins for EMSA assays.

Re: For the EMSA assay, we firstly tried to purify FgSR-His via the *E. coli* expression system, but failed to get full length recombinant FgSR protein because that FgSR-His was mainly present the inclusion body (Image a below). Later, we tried to purify FgSR^{N1-158} containing the ZF and ML domains via the *E. coli* expression system, and we obtained the recombinant FgSR^{N1-158} protein (Image b below). We performed EMSA using probe DNAs labeled with or without biotin at the 3' end of the reverse complementary chains of *cis*-element. As shown in Fig. 7g, FgSR^{N1-158} binds to the *cis*-element identified in this study.

Reviewer #3 (Remarks to the Author):

The authors provide a revised version of a previous manuscript and addressed all concerns, made in the reviews for the initial manuscript.

1. I appreciate that major concerns regarding phosphorylation sites of the transcription factor were addressed. In particular, the predicted phosphorylation sites were identified using NetPhos. And this explains in part the follow-up experiments. (see also the critics of reviewer #2, point 2)

2. Another concern was related to the identified cis-element that putatively was bound by transcription factor FgSR. Although the authors gave a lot of significant answers, concerning this cis-element, an important experiment is still missing. They were unable to demonstrate the binding of the cis-element by FgSR using an EMSA assay (see reviewer #2, point 3 and the corresponding answer).

Re: Thank you very much for your great comments. Please see responses to the comments raised by reviewer #2 above.

REVIEWERS' COMMENTS:

Reviewer #2 (Remarks to the Author):

In this revised manuscript, the authors provided direct evidence on the binding of FgSR to its binding site identified in this study. The authors also showed that FgSR directly interacted with FgHog1. In addition, they generated the S/T to A or D mutant alleles of FgSR at five predicted phosphorylation sites.

The authors presented convincing data on the regulatory role of FgSR on sterol biosynthesis in *F. graminearum*, which is quite novel by itself to this reviewer.

Regarding the model on the phosphorylation of FgSR by FgHog1 and its recruiting the SWI/SNF complex, the authors need to revise the model and conclusions based on their data. The FgSR mutant has much more severe growth defects and sensitivities to tebuconazole. Also, FgSR was still phosphorylated in the FgHog1 mutant although to a reduced level. Other kinases must also be involved in its activation.

Whereas I don't want to be overly critical, to this reviewer, some of the data presented in earlier submissions of this manuscript appeared to be contradictory to the current version and somewhat questionable. In earlier versions, the authors focused on the interaction of FgSsk2 with FgSR. In the revised manuscript, the authors showed data FgHog1 interacted with and phosphorylated FgSR. FgSsk2 but not FgHog1 was identified by yeast two-hybrid screens. However, only Fghog1-FgSR but not Ssk2-FgSR interaction was detected by BiFC assays. How? Will components of the FgHog1 MAP kinase pathway may form a protein complex in *F. graminearum* but not in yeast? Some discussion may be helpful.

Will expression of FgSR S/T-D (activation mimic or dominant active) allele be suppressive to the Fghog1 mutant? That may present the most direct evidence.

Based on studies in other fungi, increased sensitivities of the Fghog1 mutant to tebuconazole may be partly due to its role in regulating stress responses and the expression of MDR transporters. A possible discussion point.

Reviewer #3 (Remarks to the Author):

The authors addressed my major concern, the binding of FgSR to the identified cis-element. In their revision, the authors have tried to produce the FgSR protein in *E. coli*. Finally, they succeeded by using a truncated version of the gene. With this, EMSA experiments were successfully conducted and demonstrated in Fig. 7g. This includes several important controls.

I congratulate the authors for their successful experiment

REVIEWERS' COMMENTS:

Reviewer #2 (Remarks to the Author):

In this revised manuscript, the authors provided direct evidence on the binding of FgSR to its binding site identified in this study. The authors also showed that FgSR directly interacted with FgHog1. In addition, they generated the S/T to A or D mutant alleles of FgSR at five predicted phosphorylation sites. The authors presented convincing data on the regulatory role of FgSR on sterol biosynthesis in *F. graminearum*, which is quite novel by itself to this reviewer.

Re: Thank you very much for your comments and suggestions.

Regarding the model on the phosphorylation of FgSR by FgHog1 and its recruiting the SWI/SNF complex, the authors need to revise the model and conclusions based on their data. The FgSR mutant has much more severe growth defects and sensitivities to tebuconazole. Also, FgSR was still phosphorylated in the FgHog1 mutant although to a reduced level. Other kinases must also be involved in its activation.

Re: Thanks for your important suggestions. We agree with this point. The involvement of other kinase(s) in activating FgSR is added in the model (lines 515-520).

Whereas I don't want to be overly critical, to this reviewer, some of the data presented in earlier submissions of this manuscript appeared to be contradictory to the current version and somewhat questionable. In earlier versions, the authors focused on the interaction of FgSsk2 with FgSR. In the revised manuscript, the authors showed data FgHog1 interacted with and phosphorylated FgSR. FgSsk2 but not FgHog1 was identified by yeast two-hybrid screens. However, only Fghog1-FgSR but not Ssk2-FgSR interaction was detected by BiFC assays. How? Will components of the FgHog1 MAP kinase pathway may form a protein complex in *F. graminearum* but not in yeast? Some discussion may be helpful.

Re: Thanks for your insightful comments. In the earlier submission, the association between FgSR and FgSsk2 was confirmed by the Y2H and Co-IP assays, but the direct interaction between FgSR and FgSsk2 was not confirmed by BiFC. According to your valuable comments, we tested the relationship of the downstream kinase Fghog1 and FgSR during the revision. We found that FgSR interacted with FgHog1 in the Co-IP and BiFC assays, and this interaction was dependent on FgHog1 phosphorylation mediated by FgSsk2 and enhanced by tebuconazole treatment. Importantly, FgSR directly interacted with FgHog1 in the pull down assay, and was phosphorylated by FgHog1 *in vitro*. These results indicate that tebuconazole treatment activates the HOG pathway, and the activated FgHog1 further phosphorylates FgSR to regulate *FgCYP51A* expression. Thank you for your reminder, we stated that the interaction between FgSR and FgSsk2 is likely indirect, probably via the HOG pathway (lines 243-245).

In the budding yeast, Ssk2 interacts with Pbs2 and Pbs2 interacts with Hog1 both

in *in vivo* and in *in vitro* by the Co-IP and GST-pull down assays (Tatebayashi et al, 2003; Murakami et al., 2008). The current results found that FgHog1 directly interacts with FgSR and the interaction of FgSsk2 and FgSR is likely indirect, probably via the HOG pathway, which cannot support the conclusion that the three components of the HOG pathway form a complex in *F. graminearum*. It would be a very interesting point to elucidate physical relationships of these components in the near future. We discuss this point in lines 241-245.

References:

- Tatebayashi, K., Takekawa, M., & Saito, H. (2003). A docking site determining specificity of Pbs2 MAPKK for Ssk2/Ssk22 MAPKKs in the yeast HOG pathway. *EMBO J.*, 22(14), 3624-3634.
- Murakami, Y., Tatebayashi, K., & Saito, H. (2008). Two adjacent docking sites in the yeast Hog1 mitogen-activated protein (MAP) kinase differentially interact with the Pbs2 MAP kinase kinase and the Ptp2 protein tyrosine phosphatase. *Mol. Cell. Biol.*, 28(7), 2481-2494.

Will expression of FgSR S/T-D (activation mimic or dominant active) allele be suppressive to the Fghog1 mutant? That may present the most direct evidence.

Re: Thank you very much for this constructive suggestion. One previous study has showed that FgHog1 targets the transcription factor Atf1 (Nguyen et al, 2012). This study reports a novel transcription factor FgSR targeted by FgHog1. It is reasonable for us to propose that mimic activation of FgSR might only restore the defects of FgHog1 mutant partially, but not fully.

Reference:

- Nguyen TV, Schafer W, Bormann J (2012) The stress-activated protein kinase FgOS-2 is a key regulator in the life cycle of the cereal pathogen *Fusarium graminearum*. *Mol. Plant-Microbe Interact.*, 25: 1142–1156.

Based on studies in other fungi, increased sensitivities of the Fghog1 mutant to tebuconazole may be partly due to its role in regulating stress responses and the expression of MDR transporters. A possible discussion point.

Re: Thank you very much for this comment. In *Schizosaccharomyces pombe* (Kim et al, 2011; Hu et al, 2015), *Saccharomyces cerevisiae* (Montañés et al, 2011), *Cryptococcus neoformans* (Kim et al, 2011; Ko et al, 2009) and *Candida albicans* (Cohen et al, 2014), Hog1 homologs suppressed the expression of ergosterol related genes, subsequently the Hog1 mutants showed reduced sensitivity to the azole compounds. We discuss this point in lines 496-500.

In *F. graminearum*, ATP-binding cassette (ABC) transporters do not play a critical role in regulating azole sensitivity (Abou Ammar et al, 2013), which suggests that FgHog1 regulates tebuconazole sensitivity mainly via FgSR, but not likely ABC transporters.

References:

- Hu, L., Fang, Y., Hayafuji, T., Ma, Y., & Furuyashiki, T. (2015). Azoles activate Atf1 - mediated transcription through MAP kinase pathway for antifungal effects in fission yeast. *Genes Cells*, 20(9), 695-705.
- Cohen, B. E. (2014). Functional linkage between genes that regulate osmotic stress responses and multidrug resistance transporters: challenges and opportunities for antibiotic discovery. *Antimicrob. Agents Ch.*, 58(2), 640-646.
- Kim, S. Y., Ko, Y. J., Jung, K. W., Strain, A., Nielsen, K., Azoles activate Atf1 - mediated transcription through MAP kinase pathway for antifungal effects in fission yeast. *Genes Cells*, 20(9), 695-705.
- Kim, S. Y., Ko, Y. J., Jung, K. W., Strain, A., Nielsen, K., & Bahn, Y. S. (2011). Hrk1 plays both Hog1-dependent and-independent roles in controlling stress response and antifungal drug resistance in *Cryptococcus neoformans*. *PLoS ONE*, 6(4), e18769.
- Ko, Y. J., Yu, Y. M., Kim, G. B., Lee, G. W., Maeng, P. J., Kim, S., ... & Bahn, Y. S. (2009). Remodeling of global transcription patterns of *Cryptococcus neoformans* genes mediated by the stress-activated HOG signaling pathways. *Eukaryot. cell*, 8(8), 1197-1217.
- Montañés, F. M., Pascual - Ahuir, A., & Proft, M. (2011). Repression of ergosterol biosynthesis is essential for stress resistance and is mediated by the Hog1 MAP kinase and the Mot3 and Rox1 transcription factors. *Mol. Microbiol.*, 79(4), 1008-1023.
- Abou Ammar G, Tryono R, Doñil K, Karlovsky P, Deising HB, et al. (2013). Identification of ABC transporter genes of *Fusarium graminearum* with roles in azole tolerance and/or virulence. *PLoS ONE* 8(11): e79042.

Reviewer #3 (Remarks to the Author):

The authors addressed my major concern, the binding of FgSR to the identified cis-element.

In their revision, the authors have tried to produce the FgSR protein in E. coli. Finally, they succeeded by using a truncated version of the gene. With this, EMSA experiments were successfully conducted and demonstrated in Fig. 7g. This includes several important controls.

I congratulate the authors for their successful experiment

Re: Thanks for your positive comments on the manuscript.